

# Proliferating activity in a bryozoan lophophore

Natalia Shunatova[1] and Ilya Borisenko[2]

[1] Department of Invertebrate Zoology, Biological Faculty, St. Petersburg State University, St. Petersburg, Russia
[2] Department of Embryology, Biological Faculty, St. Petersburg State University, St. Petersburg, Russia

## ABSTRACT

Bryozoans are small benthic colonial animals; their colonies consist of zooids which are composed of a cystid and polypide. According to morphological and molecular data, three classes of bryozoans are recognized: Phylactolaemata, Gymnolaemata and Stenolaemata. Bryozoans are active suspension feeders and their feeding apparatus, the lophophore, is fringed with a single row of ciliated tentacles. In gymnolaemates, the lophophore is bell-shaped and its tentacles may be equal in length (equitentacled lophophores) or some tentacles may be longer than others (obliquely truncated lophophores). In encrusting colonies, polypides with obliquely truncated lophophores usually border specific sites of excurrent water outlets (colony periphery and chimneys) where depleted water has to be removed. It is known that during colony astogeny, colony-wide water currents rearrange: new chimneys are formed and/or location of the chimneys within a given colony changes with time. Such rearrangement requires remodeling of the lophophore shape and lengthening of some tentacles in polypides surrounding water outlets. However, proliferating activity has not been described for bryozoans. Here, we compared the distribution of S-phase and mitotic cells in young and adult polypides in three species of Gymnolaemata. We tested the hypothesis that tentacle growth/elongation is intercalary and cell proliferation takes place somewhere at the lophophore base because such pattern does not interfere with the feeding process. We also present a detailed description of ultrastructure of two parts of the lophophore base: the oral region and ciliated pits, and uncover the possible function of the latter. The presence of stem cells within the ciliated pits and the oral region of polypides provide evidence that both sites participate in tentacle elongation. This confirms the suggested hypothesis about intercalary tentacle growth which provides a potential to alter a lophophore shape in adult polypides according to rearrangement of colony wide water currents during colony astogeny. For the first time deuterosome-like structures were revealed during kinetosome biogenesis in the prospective multiciliated epithelial cells in invertebrates. Tentacle regeneration experiments in *Electra pilosa* demonstrated that among all epidermal cell types, only non-ciliated cells at the abfrontal tentacle surface are responsible for wound healing. Ciliated cells on the frontal and lateral tentacle surfaces are specialized and unable to proliferate, not even under wound healing. Tentacle regeneration in *E. pilosa* is very slow and similar to the morphallaxis type. We suggest that damaged tentacles recover their length by a mechanism similar to normal growth, powered by proliferation of cells both within ciliated pits and the oral region.

Corresponding author
Natalia Shunatova,
n.shunatova@spbu.ru

## INTRODUCTION

Cell proliferation is the main contributor to the growth of an organism and also participates in maintenance of its body during the lifespan. In adults, proliferating cells contribute to physiological cell turnover and replacement of injured cells and tissues. Different aspects of cell proliferation during morphogenesis and regeneration were described for various animals: sponges (*Alexander et al., 2014*; *Borisenko et al., 2015*; *Ereskovsky et al., 2015*; *Kahn & Leys, 2016*), cnidarians (*Chera et al., 2009*; *Fritz et al., 2013*; *Gahan et al., 2016*), plathyhelminths (*Reddien & Alvarado, 2004*; *Saló et al., 2009*; *Cebrià, Adell & Saló, 2018*), annelids (*Fischer & Dorresteijn, 2004*; *Paulus & Müller, 2006*; *Brinkmann & Wanninger, 2010*; *Grdisa, 2010*; *Yoshida-Noro & Tochinai, 2010*; *Bely, 2014*), insects (*Milán, Campuzano & García-Bellido, 1996*; *Neufeld et al., 1998*; *Baonza & Freeman, 2005*), mollusks (*Grimaldi et al., 2004*; *Henry, Okusu & Martindale, 2004*; *Redl et al., 2016*), vertebrates (*Hay & Fischman, 1961*; *Bettencourt-Dias, Mittnacht & Brockes, 2003*; *Warren, Puskarczyk & Chapman, 2009*), etc. However, many bilaterian taxa, including bryozoans, are still unexplored in this respect.

Bryozoans are small benthic animals which are almost exclusively colonial (*Schwaha et al., 2019*). Their colonies consist of zooids, which are composed of a cystid (immovable part, often calcified) and polypide (movable soft tissues: a lophophore, tentacle sheath, gut and associated muscles, and nervous system). According to morphological and molecular data, three classes of bryozoans are recognized: Phylactolaemata, Gymnolaemata and Stenolaemata (*Mukai, Terakado & Reed, 1997*; *Waeschenbach, Taylor & Littlewood, 2012*). Gymnolaemates are the most diverse and abundant extant bryozoan group. In all groups of bryozoans polypides are short lived and degenerate after a few weeks with a formation of a residual mass, or brown body. Cystids survive much longer even in Phylactolaemata: polypides in this group never regenerate. In the majority of gymnolaemates and stenolaemates a polypide regenerates within the same cystid. The general overview of this process was provided for several species at light microscopic level (see *Mukai, Terakado & Reed, 1997* for a review). Such regular degeneration-regeneration cycles require the presence of stem cells within each zooid.

Bryozoans are active suspension feeders (*Jørgensen, 1966*) and their feeding apparatus, a lophophore, is fringed with a single row of ciliated tentacles. Tentacle ultrastructure has been described for three species of Gymnolaemata (*Lutaud, 1973*; *Smith, 1973*; *Gordon, 1974*; *Mukai, Terakado & Reed, 1997*). The outer tentacle epithelium includes both non-ciliated and ciliated cells; the latter are arranged in five longitudinal rows and, correspondingly, form five ciliary bands: a single frontal band and paired latero-frontal and lateral bands. Functions of ciliary bands are well understood: they create a water current and participate in the capture of food. The tentacle epidermis rests on a thick layer of extracellular matrix (ECM), which is continuous with that of the lophophore base. Each tentacle houses a coelomic cavity which is an outgrowth of the lophophore coelom.

The coelomic lining of the tentacles includes both epithelial cells on the lateral surfaces and myoepithelial cells on the frontal and abfrontal sides. Myoepithelial cells form the frontal and abfrontal tentacle muscles. Within a lophophore, tentacles may be equal in length (equitentacled lophophores) or some tentacles may be longer than the other (obliquely truncated lophophores).

The tentacles unite at their bases and join the lophophore base which is very complex in structure. At this region, the tentacle coeloms join the ring coelomic canal surrounding the mouth. There is a single description of the microanatomy of the lophophore base in *Cryptosula pallasiana* (Moll, 1803) by *Gordon (1974)*. He found a specific structure between tentacle bases and termed them "ciliated pits". The ciliated pits are small structures (about three μm in diameter and 25–30 μm deep), and their upper two thirds are ciliated. A similar structure was reported by *Schwaha & Wood (2011)* for a ctenostome *Hislopia malayensis* Annandale, 1916. Unfortunately, in both cases the authors provided no further details on their structure and mentioned that the possible function of the ciliated pits is unknown.

During feeding, the tentacle ciliation is responsible for creating water currents bringing food to the lophophore and participates in particle retention and transport. Food-depleted water leaves the lophophore between the tentacles and has to be removed from the colony. Different variants of colony-wide water currents were described for bryozoans. Among them, the most specific way of the water removal in encrusting colonies is a formation of excurrent water outlets, or chimneys, which were first described for large colonies of *Membranipora membranacea* (Linnaeus, 1767) (*Banta, McKinney & Zimmer, 1974*). Several types of chimneys are recognized, and there is a vast literature describing them. In many cases, the chimneys are surrounded by the polypides with obliquely truncated lophophores, and their longest tentacles border the chimney (*Cook, 1977*; *Winston, 1978*, *1979*; *Cook & Chimonides, 1980*; *Lidgard, 1981*; *Dick, 1987*; *McKinney, 1990*). The rest of the polypides in the colony usually have equitentacled lophophores. Polypides with obliquely truncated lophophores are also located at the colony periphery, and their longest tentacles face the colony edge. During colony astogeny, either new chimneys are formed, and/or the location of the chimneys within the given colony changes with time (*Von Dassow, 2005a*, *2005b*, *2006*). In many cases, this happens during the same degeneration-regeneration cycle. Thus, the question arises: are the polypides surrounding the new chimney capable of lengthening some of their tentacles and changing the shape of their lophophores? For two cheilostomes *Holoporella brunnea* (=*Celleporaria brunnea* (Hincks, 1884)) and *Membranipora serrilamella* (=*M. villosa* Hincks, 1880), *Dick (1987)* mentioned the possibility of a transformation from obliquely truncated lophophore to equitentacled one, and vice versa. He suggested that the reason for this transformation is the lophophore position respective to the changing excurrent flow.

Considering data reported by *Dick (1987)*, one can suggest that such an elongation of the tentacles implies the presence of proliferating cells either in the tentacle itself or at the lophophore base. Proliferating activity within the lophophore has not been described for bryozoans. Nevertheless, the presence of blastemic cells was mentioned within the oral

region of the polypide in *C. pallasiana* (*Gordon, 1974*) and close to the ganglion of a degenerating female polypide in *Alcyonidium polyoum* (Hassall, 1841) (*Matricon, 1963*).

It is well known that different benthic animals prey on bryozoans using different mechanisms, and usually they consume a whole polypide or a considerable part of it (*Iyengar & Harvell, 2002*; *Berning, 2007*; *Lidgard, 2008*; *Lindsay, 2010*). However, there is a single report that individual tentacles could be bitten off by a predator: *Marcus (1941)* described that chironomid larvae bite off the tips of tentacles in polypides of the genus *Stolella* (Phylactolaemata). He also pointed out that such bitten tentacles regenerate. The experimental regeneration of tentacles was studied only in phylactolaemates, and it was demonstrated that tentacles, when cut off, regenerate in the course of several days (*Otto, 1921*; *Oda, 1954*). Unfortunately, both authors did not provide any data on the location of proliferating cells. Similar data on gymnolaemates and stenolaemates are absent. Nevertheless, based on the idea that tentacles can lengthen in adult polypides, it can be assumed that polypides have the ability to regenerate damaged tentacles.

Here we applied EdU-labeling technique to bryozoans to study proliferating activity within the lophophore. For three species of gymnolaemates with obliquely truncated lophophores, we tested the hypothesis that the tentacle growth/elongation is intercalary or, in other words, proliferating cells are not scattered along the tentacle but are located in a specific site(s) within the lophophore base. Such pattern makes the tentacle elongation possible without interfering with the feeding process. We present a detailed description of two regions of the lophophore base housing proliferating cells: the oral region and ciliated pits in four gymnolaemate species. We also tried to detect the regenerative capacity of polypides in *Electra pilosa* (Linnaeus, 1767) after microsurgical amputation of the distal part of the tentacles.

## MATERIALS AND METHODS

### Animals

We studied ultrastructure of the lophophore base in four species: *Rhamphostomella ovata* (Smitt, 1868); *Electra pilosa* Linnaeus, 1767; *Aquiloniella scabra* (Van Beneden, 1848) and *Eucratea loricata* (Linnaeus, 1758) (Gymnolaemata: Cheilostomatida). Proliferating activity was registered in *R. ovata*, *E. pilosa* and *A. scabra*: these species have both equitentacled and obliquely truncated lophophores. Material was collected in the vicinity of Research and Educational Station "Belomorskaya" of SPbSU (White Sea, Kandalaksha Bay, Chupa Inlet) by SCUBA diving in 2016–2017.

*Rhamphostomella ovata* and *E. pilosa* have encrusting colonies with centrifugal growth, and the youngest zooids are located at the colony periphery. *A. scabra* and *E. loricata* have erect colonies with youngest zooids located at the tips of the branches.

Different age classes of zooids were referred as followed. The youngest zooids with fully developed and feeding polypides, which are the closest to the growth margin, are assigned as the 1st age class. Zooids of the 1st age class were budded by the zooids of the 2nd age class; zooids of the 2nd age class were budded by zooids of the 3rd age class, etc. Polypides of the 1st and 2nd age classes are referred here as "young" polypides, and polypides of 4th and older age classes as "adult" polypides.

## Transmission electron microscopy and scanning electron microscopy study

Ten colonies of each species were anesthetized in the mixture of isotonic solution of magnesium chloride and seawater (ratio 1:1) and fixed in 2.5% glutaraldehyde solution in 0.1 M isotonic cacodylate buffer (supplemented with sucrose to reach 750 mOsmol). For TEM five colonies of each species were postfixed with 1% osmium tetroxide in 0.1 M cacodylate buffer (supplemented with sucrose to reach 750 mOsmol), dehydrated through a graded ethanol series, transferred through an ethanol–acetone mixture to pure acetone and then embedded in epoxy resin. Serial sections were obtained on an ultramicrotome Leica EM UC7 using diamond knife in different projections. Semithin sections were stained with toluidine blue; ultrathin sections were stained in uranyl acetate and lead citrate and then investigated with Jeol JEM 1400 electron microscope. For SEM five colonies of each species were processed according to standard protocols, critical point dried, and sputter coated with 20 nm gold. Scanning electron micrographs were made using a Tescan MIRA3 LMU electron microscope. The images were processed with Photoshop CS5.

## Proliferating activity

Proliferating activity was detected by incorporation of labeled thymidine analog, 5-ethynyl-2′-deoxyuridine (or EdU), during S-phase of cell cycle, with subsequent visualization of labeled DNA by click-reaction (*Salic & Mitchison, 2008*). EdU and the reactive dye, sulfo-Cyanine5 azide, were obtained from Lumiprobe (Russia). Five colonies of each species were placed in filtered sea water with 100 or 300 µM EdU, diluted from 250 mM stock in DMSO. Animals were incubated during 3 and 6 h at 7 °C in Petri dishes. After that, colonies were anesthetized as described above and fixed in 4% PFA in 0.1M PBS (pH 7.4) overnight. Some polypides remained retracted during anesthetization; in such cases we dissolved a calcified layer of ectocyst with 5% EDTA in 0.1M PBS before staining. Fixed colonies were washed in 0.1M PBS with addition of 0.1% Tween-20 three times (10 min each) and permeabilized with 0.5% Triton X-100 in 0.1M PBS during 15 min on a shaker. After rinsing in PBS, colonies were incubated in staining mix (Tris-HCl pH 8.5 100 mM, $CuSO_4$ 4 mM, sulfo-Cyanine5 azide 2 µM, ascorbic acid 50 mM) during 1 h at room temperature in the dark. Then colonies were rinsed 0.1M PBS with addition of 0.1% Tween-20 two times (10 min each) and blocked in 5% sheep serum in 0.1M PBS with addition of 0.1% Tween-20 during 1 h on a shaker. Nuclei were counterstained with DAPI at 1 µg/ml. Polypides were extracted with fine needles under stereomicroscope and mounted in glycerol with DABCO addition as an antifade. Specimens were examined using the confocal laser scanning microscope Leica TCS SPE. The images were processed by ImageJ software (FiJi).

## Regeneration experiments

Regeneration experiments were made only for *Electra pilosa*. The most probable damage not followed by zooid death is biting off the distal part of tentacle(s) by a predator. We chose 10 large colonies including about 350–400 zooids and anesthetized them

(as described above). In each colony we selected 10 polypides belonging to the 5th and 6th age classes and amputated the distal part (from 1/4 to 1/2) of 2–4 tentacles. After that, colonies were washed in seawater to remove anesthetic and were kept in seawater in isothermic room at 7 °C. Colonies were fixed after 24, 48, 72, 120 and 148 h surgery (two colonies each time), and 6 h prior fixation colonies were moved to the mixture of EdU (final concentration of 300 µM) and isotonic solution of magnesium chloride. After labeling, colonies were fixed in 4% PFA in 0.1M PBS and processed as described above.

## RESULTS

### General remarks on tentacle structure

Polypides of studied species possess different number of tentacles: 10 in *Eucratea loricata*, 12–16 in *Electra pilosa*, 14–17 in *Rhamphostomella ovata* and *Aquiloniella scabra*. General view of feeding polypides is given in Fig. 1A. In all studied species, the tentacles are covered with a one-layered epithelium; its cells are arranged in nine continuous and one (the tenth) intermittent rows (Fig. 1B). Following *Smith (1973)*, we used an alphabetical code to refer to the different cell types (Fig. 1C). The frontal side of the tentacle is covered with a single row of A-cells (frontal cells). Paired rows of latero-frontal B-cells are located on either side of the frontal A-cells. Paired fronto-lateral (C-cells) and abfronto-lateral (D-cells) rows cover the lateral surfaces of the tentacle. The abfrontal surface of the tentacle is composed of paired rows of abfrontal E-cells. Between them, at the midline of the abfrontal surface, there is a single intermittent row of F-cells.

Each tentacle possesses five longitudinal ciliary bands described for Bryozoa. The frontal band is formed by cilia of multiciliated A-cells, the latero-frontal bands are composed of cilia of monociliated B-cells, and the lateral bands includes cilia of multiciliated C-and D-cells (Fig. 1D). The abfrontal tentacle surface bears regularly alternated multi- and monociliated putative sensory F-cells (Fig. 2A), and each tentacle tip has a terminal tuft of several cilia belonging to 3–4 apical cells. The alternating arrangement of mono-and multiciliated F-cells was similar to that described before (*Shunatova & Nielsen, 2002*). Thus, in "adult" polypides of all studied species, both mono- and multiciliated F-cells were more or less evenly spaced along the entire tentacle length, with an average distance between them of about 10–15 µm (Figs. 2A and 2B). However, at the tentacle bases of young polypides (1st and 2nd age classes), the distance between F-cells is much shorter than that in the distal and middle parts of the same tentacle (Fig. 2C).

The tentacle epithelium is supported with a prominent ECM layer which forms a hollow tube with two longitudinal abfronto-lateral ridges, or horns. Fine structure of the cells of the tentacle coelomic lining has been recently described in details (*Shunatova & Tamberg, 2019*).

Along the entire tentacle length, cell composition of tentacles in studied species is identical.

### Lophopore base

Lophophore base is very complex in structure and here we address only two sites: ciliated pits and the region surrounding the mouth. The structure of each region is the same in

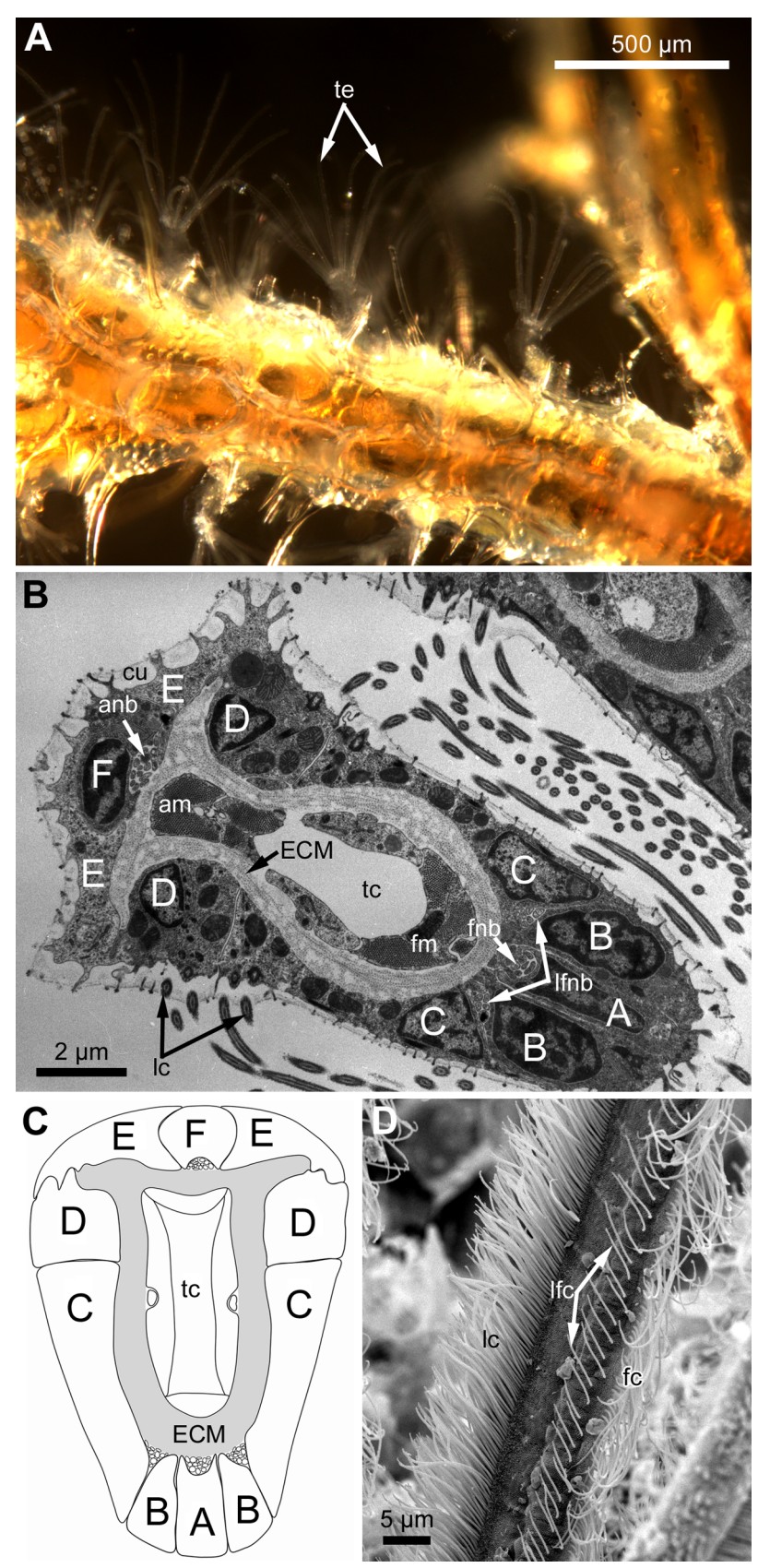

**Figure 1 General structure of tentacles in gymnolaemates.** (A) Feeding colony of *Electra pilosa* (in life). Note the funnel shape of lophophores. (B) Cross-section of the tentacle in *Eucratea loricata* (TEM; the frontal surface of the tentacle to the bottom right). (C) A schematic drawing of tentacle cross-section demonstrating an alphabetical code referring to the different cell types (the frontal surface of the tentacle to the bottom). (D) Tentacle ciliation in *Aquiloniella scabra* (SEM; lateral view of the tentacle) showing lateral, latero-frontal and frontal ciliary bands. Abbreviations: A, frontal cell (A-cell); B, latero-frontal cell (B-cell); C, fronto-lateral cell (C-cell); D, abfronto-lateral cell (D-cell); E, abfrontal cell (E-cell); F, putative sensory cell (F-cell) forming intermittent row at the midline of the abfrontal tentacle surface; anb, abfrontal nerve bundle; cu, cuticle; ECM, extracellular matrix; fc, frontal ciliary band; fnb, frontal nerve bundle; lc, lateral ciliary band; lfc, latero-frontal ciliary band; lfnb, latero-frontal nerve bundle; tc, lumen of the tentacle coelom.

young and adult polypides (except for the number of the cells undergoing mitosis—see below).

### Ciliated pits

At the proximal end of the tentacle, the number of E-cells increases from two to three, and they combine both with the E-cells of the adjoining tentacles and with the epidermis of the tentacle sheath (Figs. 3A and 3B). At this level, the abfronto-lateral horns of the tentacle ECM change their shape: it becomes irregular conical in cross sections. Between them, two additional small projections of the tentacle ECM appear; they are located between the bases of E-cells (Figs. 3A and 3B). Each of the four projections gives rise to a thin outgrowth of ECM, and together they envelope the basal portions of E-cells and combine with the ECM of the tentacle sheath (Figs. 3B and 3C). These outgrowths also stretch basally and unite with the ECM of the septum dividing the lophophore and main body cavity (Fig. 3C). At this level, B- and C-cells of adjacent tentacles contact each other (Fig. 3D) and isolate the lumen of the ciliated pit.

The ciliated pits are blind-ended conical invaginations of the epidermis between the tentacle bases (Figs. 3E and 3F). The number of the ciliated pits corresponds to the tentacle number. The opening of the ciliated pit is surrounded with C-and D-cells of adjacent tentacles and a single E-cell which shifted from either of the tentacles; the latter takes the abfrontal side of the ciliated pit. Basally, these five cells are followed by five vertical cell rows composing the walls of the ciliated pit, and here we use the following designation for them. Rows of CP-and DP-cells are located below the corresponding C-and D-cells of the neighboring tentacles and a single row of EP-cells occupies the abfrontal surface of the ciliated pit (Fig. 4). The depth of the ciliated pits varies across studied species. The deepest ciliated pits were found in *R. ovata* (about 30 μm): they include up to 11 cells in the CP-and DP-rows and up to seven cells in the EP-row. Polypides of *E. loricata* have the shortest ciliated pits (about 10 μm): their CP-and DP-rows are composed of three cells, and EP-row includes only two cells. Three regions are recognized along the axis of the ciliated pit: distal, middle and proximal (Figs. 3F and 4).

The distal part of the ciliated pit houses a prominent funnel-shaped lumen that opens into the environment. The fine structure of both CP-and DP-cells in general resembles that of C-and D-cells, although the shape of the cells change to an irregular trapezoid and their apical surface is limited by the lumen of the ciliated pit (Figs. 5A and 5B).

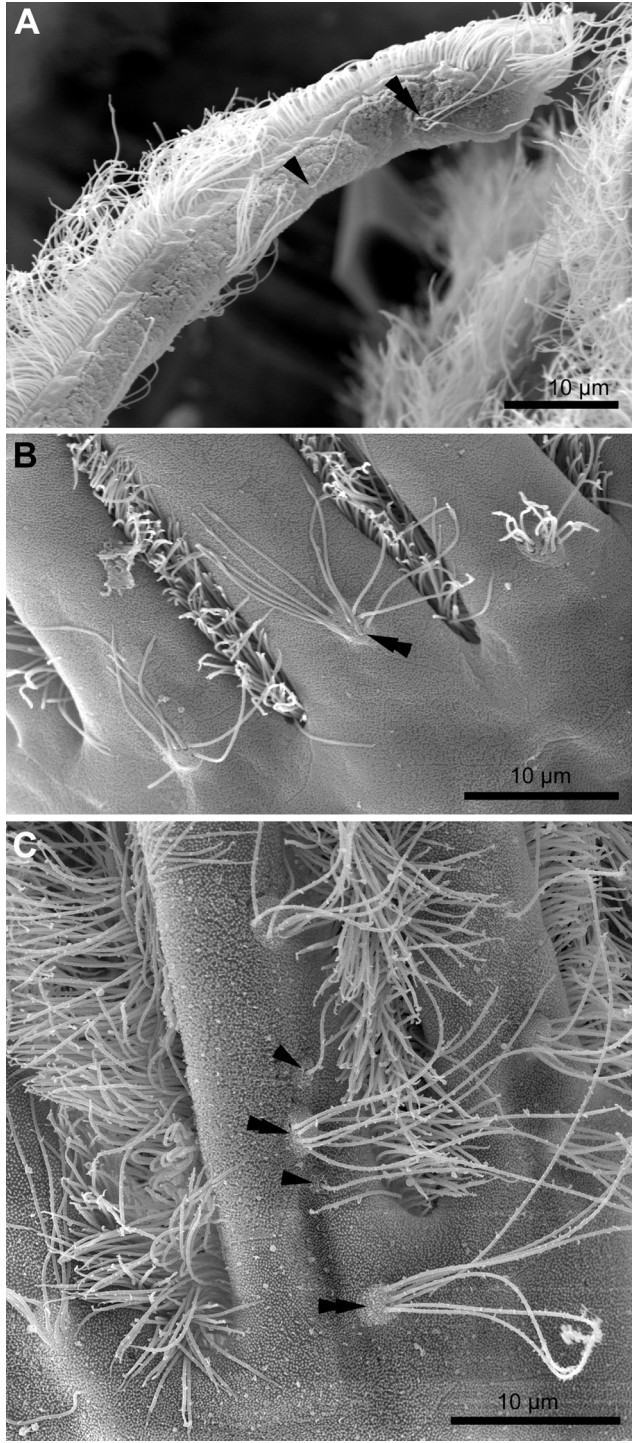

**Figure 2 Arrangement of mono-and multiciliated F-cells in young (2nd age class) and adult (5th age class) polypides of *Aquiloniella scabra*.** (A) The distal part of the tentacle in adult polypide (SEM; abfrontal view). Note the regular arrangement of mono-(arrowhead) and multiciliated (double arrowhead) F-cells. (B) The basal part of the tentacle in adult polypide (SEM; abfrontal view). Note a single multiciliated F-cell at the tentacle base. (C) The basal part of the tentacle in young polypide (SEM; abfrontal view). Note the tightly packed mono-(arrowheads) and multiciliated (double arrowheads) F-cells at the tentacle base.

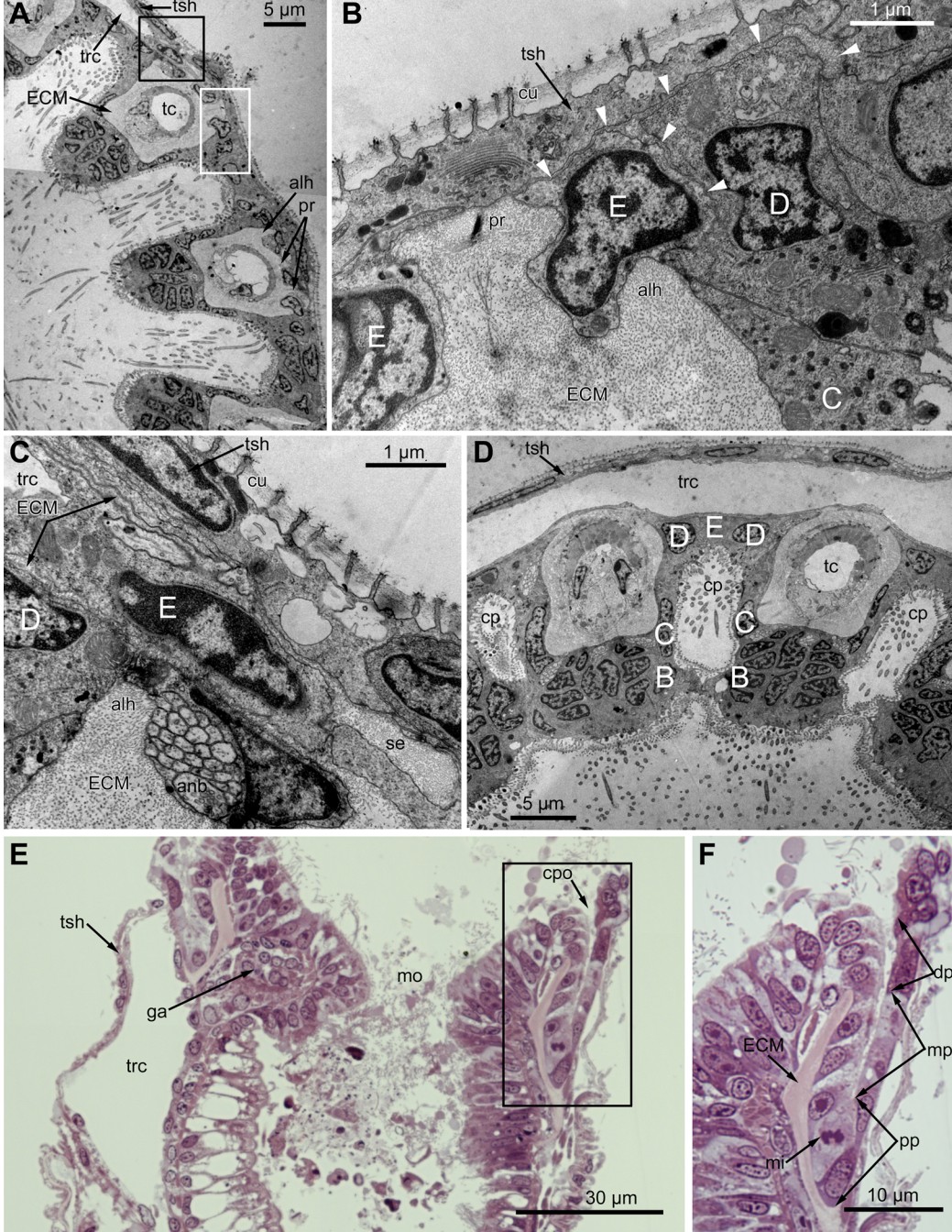

**Figure 3 Microanatomy of the lophophore base at the level of the ciliated pits' openings.**
(A–D) Oblique-cross sections of the lophophore base of adult polypide in *Aquiloniella scabra* (TEM);
(E and F) longitudinal semi-thin section of adult polypide in *Rhamphostomella ovata*. (A) At the
proximal end of the tentacle, the number of E-cells increases from two to three, and they combine both
with the epidermis of the tentacle sheath and with the E-cells of the adjoining tentacles. The white frame
corresponds to the enlarged area in (B); the black frame corresponds to the enlarged area in (C). (B) The
enlarged area in (A), demonstrating thin projections of the tentacle ECM (arrowheads) enveloping the
basal portions of E-cells. (C) Enlargement of (A), demonstrating thin projections of the tentacle ECM
uniting with the ECM of the septum which divides the lophophore and main body cavity. (D) B-and C-cells
of adjacent tentacles contact each other and isolate the lumen of the ciliated pit. (E) Longitudinal semi-thin
section through the ciliated pit which is a blind-ended conical invagination of the epidermis. The black

**Figure 3** (continued)
frame corresponds to the enlarged area in (F). (F) The enlarged area in (E), demonstrating three regions along the axis of the ciliated pit. Note the mitotic figure in the proximal part of the ciliated pit. Abbreviations: A, frontal cell (A-cell); B, latero-frontal cell (B-cell); C, fronto-lateral cell (C-cell); D, abfronto-lateral cell (D-cell); E, abfrontal cell (E-cell); alh, abfronto-lateral horns of tentacle ECM; anb, abfrontal nerve bundle; cp, ciliated pit; cpo, opening of the ciliated pit; cu, cuticle; dp, distal part of the ciliated pit; ECM, extracellular matrix; ga, ganglion; mi, mitotic figure; mo, mouth; mp, middle part of the ciliated pit; pp, proximal part of the ciliated pit; pr, additional projections of the tentacle ECM; se, septum dividing the lophophore and main body cavity; tc, tentacle coelom; trc, main body cavity; tsh, tentacle sheath.

The number of cilia decreases towards the middle region of the ciliated pit and the regularity of their arrangement is lost (Fig. 5C). Each cilium of CP-and DP-cells possesses a single kinetosome with two cross-striated rootlets resembling that of cilia in C- and D-cells in location though lateral rootlets in CP-cells are rather short (Figs. 5A and 5B). Non-ciliated EP-cells are trapezoid in shape with irregularly shaped nuclei and the usual set of organelles (Fig. 5D). The area of the apical surface of each cell type decreases towards the bottom of the ciliated pit.

The middle part of the ciliated pit contains a very narrow tube-shaped lumen; usually there is a single cilium within it (Fig. 6A), and only two out of the ciliated pits lack cilia. This cilium arises from the CP-cell located at the bottom of the lumen; the rest of CP-and DP-cells in the middle part are non-ciliated. Sometimes, the uppermost CP-and DP-cells in this region (i.e., located at the border with the distal part of the ciliated pit) possess numerous irregularly arranged kinetosomes located within the apical part of the cells (Fig. 6B). Some CP-and DP-cells, located more basally, demonstrate different stages of kinetosome biogenesis. We found clusters of electron-dense "fibrogranular material" (Fig. 6C), vacuoles with fibrillar electron-dense or electron-lucent content (Fig. 6D), electron-dense spherical bodies (or deuterosome-like structures) surrounded with numerous procentrioles (Figs. 6A, 6C and 6E). Often, one of the procentrioles is larger and lies at the right angle to the others (Fig. 6E). Rarely, the named stages are present simultaneously in the same cell. In all cases two to three dictyosomes of Golgi complex are located close to them (Figs. 6B and 6E). At the latest stage of kinetosome biogenesis, they become separated from deuterosome-like structures (Fig. 6F), and in the apical part of the cells there are developing cross-striated rootlets surrounded by electron-dense "fibrogranular material" (Fig. 6E).

The proximal region of the ciliated pit lacks a lumen and the regularity of cell arrangement is lost (Fig. 7A); we denote the cells within this part as P-cells. P-cells are small, irregular- or rounded in shape; they have relatively large nuclei, few mitochondria, scarce small cisterns of ER, free ribosomes and rarely free kinetosomes (Figs. 7B and 7C). Some P-cells undergo mitosis (Figs. 7D and 7E).

### Oral region

In the most proximal part of the tentacle, the frontal row of A-cells is two cells wide (Fig. 8A). At the level of the openings of the ciliated pits, where C-and D-cells of adjoining tentacles combine, the rows of A-, B-and C-cells join the oral region (Fig. 8B).

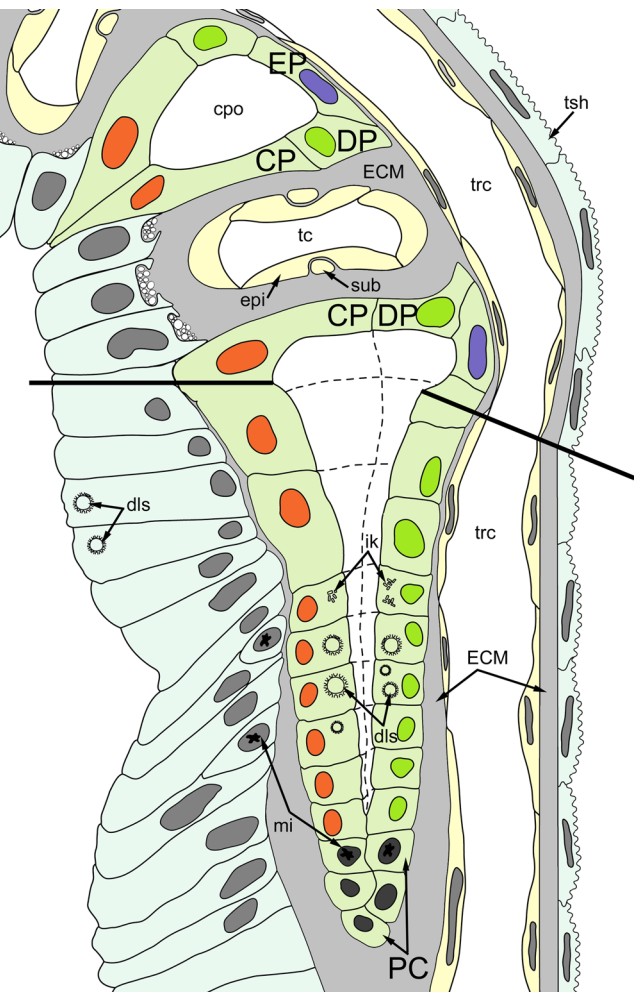

**Figure 4 A schematic drawing of the ciliated pit in *Rhamphostomella ovata*.** A sector of the lopho-phore base is shown (mouth area to the left). Tentacles are cut at the bases at the level of the opening of ciliated pits. Cilia are not shown. The cells of the ciliated pit are colored in light green; epidermal cells of the tentacle sheath and the oral region are shaded in light blue; extracellular matrix is light grey; cells of coelomic lining are light yellow. Color code for nuclei: orange for CP-cells, green for DP-cells, violet for EP cells, dark grey for P-cells, grey for the rest of the cells. Abbreviations: CP and DP, vertical rows of the cells composing the lateral walls of the ciliated pit and located below the corresponding C-and D-cells of the neighboring tentacles; EP, cells composing the abfrontal surface of the ciliated pit; PC, P-cells (cells located in the proximal region of the ciliated pit); cpo, ciliated pit opening; dls, deuterosome-like structures; ECM, extracellular matrix; epi, epiperitoneal cell lining lateral surface of the tentacle coelom; ik, irregularly arranged kinetosomes; mi, mitotic figures; sub, subperitoneal cell of the tentacle coelomic lining; tc, tentacle coelom; trc, main body cavity; tsh, tentacle sheath.

Cells forming the outer rim of the oral region lack cilia (Figs. 8C and 8D), although frontal cilia often mask it (Fig. 8B). In *E. loricata*, the apical surface of these cells has very long microvilli (Fig. 8D). The rest of the oral region is densely ciliated. The height of the cells of the oral region gradually increases towards the mouth. In *A. scabra*, *E. pilosa* and *E. loricata* all cells of the oral region are multiciliated and similar in structure. They are column-shaped with elongated or irregular shaped nuclei; their cytoplasm contains

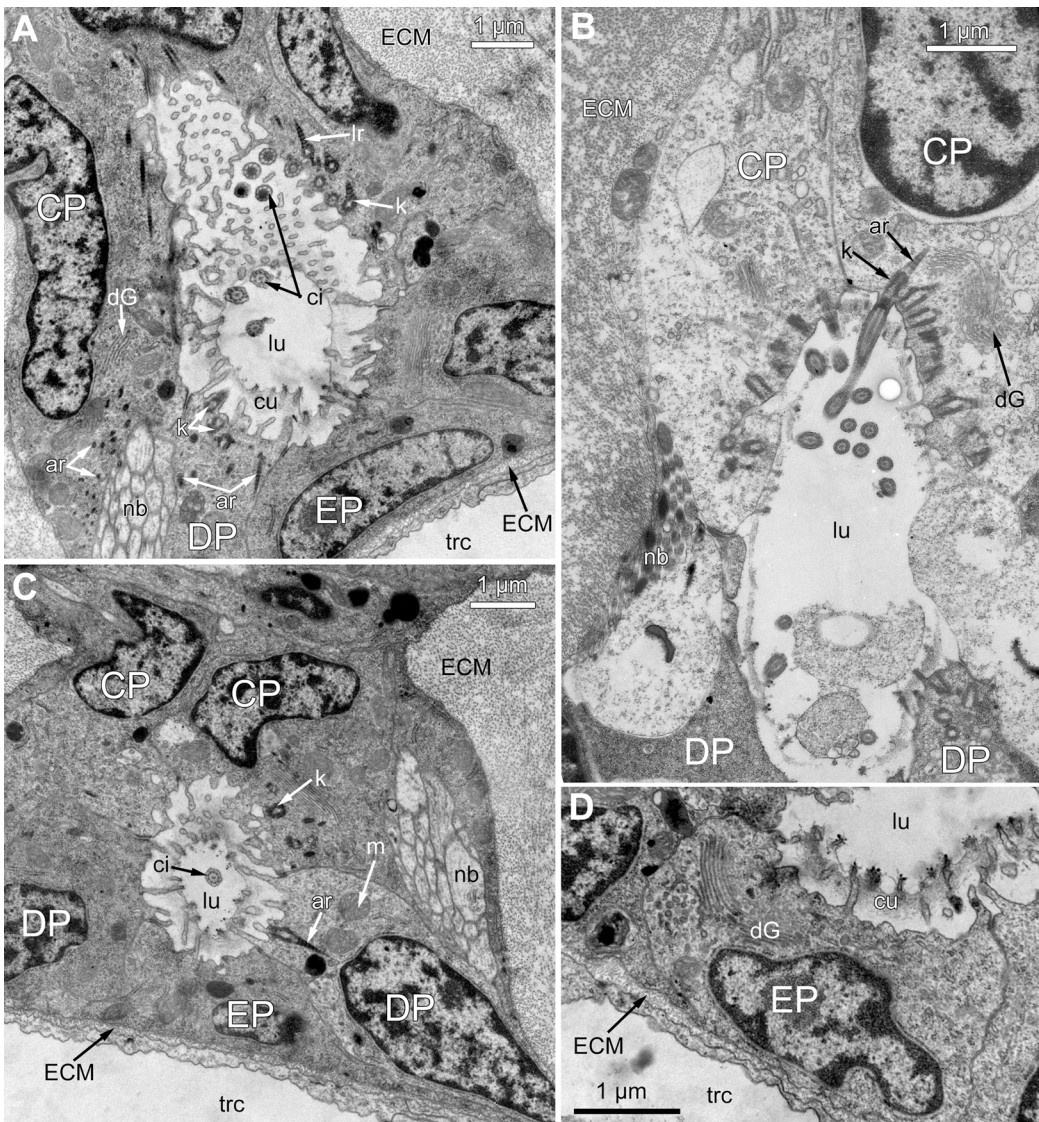

**Figure 5 Fine structure of the distal region of the ciliated pits (TEM; oblique-cross sections of the lophophore base).** (A) Details of the basal complex in CP-and DP-cells located close to the opening of the ciliated pit (*Aquiloniella scabra*; mouth upwards). (B) The fine structure of CP-cells located close to the ciliated pit opening (*Rhamphostomella ovata*; mouth upwards). (C) CP-and DP-cells located at the border with the middle region of the ciliated pit possess fewer cilia (*A. scabra*; mouth upwards). (D) The ultrastructure of EP-cells (*A. scabra*; mouth upwards). Abbreviations: CP and DP, vertical rows of the cells composing the lateral walls of the ciliated pit and located below the corresponding C-and D-cells of the neighboring tentacles; EP, cells composing the abfrontal surface of the ciliated pit; ar, axial rootlet; ci, cilium; cu, cuticle; dG, dictyosome of Goldgi complex; ECM, extracellular matrix; k, kinetosome; lr, lateral rootlet; lu, lumen of the ciliated pit; m, mitochondrion; nb, nerve bundle; trc, main body cavity.                           

numerous small membrane-bounded vesicles with electron-lucent content (Fig. 8E). Each cilium has a kinetosome with two cross-striated rootlets.

In *R. ovata*, the peripheral part of the oral region (situated close to the outer rim at the level of the middle region of the ciliated pits) is composed of two cell types. Cells located in the continuation of the rows of A-cells have an electron-dense cytoplasm, and their

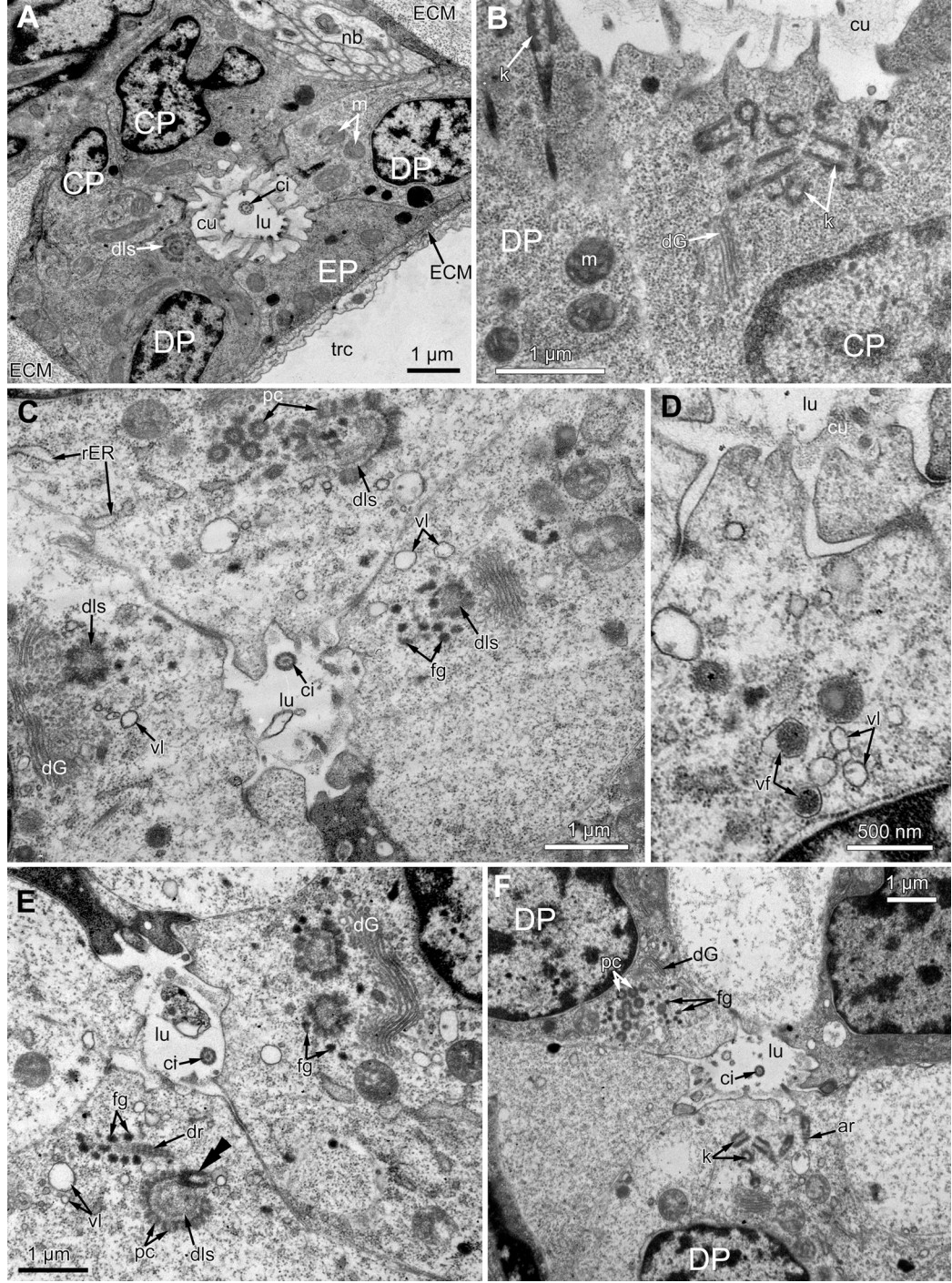

**Figure 6 The fine structure of the middle region of the ciliated pits (TEM; oblique cross-sections of the lophophore base in polypide of 3rd age class).** (A) There is a single cilium within the lumen of the ciliated pit (*Aquiloniella scabra*; mouth left upwards). Note the deuterosome-like structure in the DP-cell. (B) The apical part of CP-cell contains numerous irregularly arranged kinetosomes (*Eucratea loricata*; mouth to the right). (C–F): different stages if kinetosome biogenesis in polypide of *Rhamhostomella ovata*. (C) Clusters of electron-dense "fibrogranular material" and deuterosome-like structures in the apical parts of both CP-and DP-cells (mouth upwards). (D) The apical part of DP-cell contains vacuoles with fibrillar electron-dense or electron-lucent content (mouth upwards). (E) Deuterosome-like structures and developing cross-striated rootlets surrounded by electron-dense "fibrogranular material"

**Figure 6 (continued)**
(mouth to the right). Note that one of the procentrioles (double arrowhead) is larger and lies at the right angle to the rest. (F) Late stages of kinetosome biogenesis in DP-cells: one cell contains free procentrioles and "fibrogranular material", the other has irregularly arranged kinetosomes, some of them with axial rootlet (mouth to the right). Abbreviations: CP and DP, vertical rows of the cells composing the lateral walls of the ciliated pit and located below the corresponding C-and D-cells of the neighboring tentacles; EP, cells composing the abfrontal surface of the ciliated pit; ar, axial rootlet; ci, cilium; cu, cuticle; dG, dictyosome of Goldgi complex; dls, deuterosome-like structure; dr, developing rootlet; ECM, extracellular matrix; fg, "fibrogranular material", k, kinetosome; lu, lumen of the ciliated pit; m, mitochondrion; nb, nerve bundle; pc, procentrioles; rER, rough endoplasmic reticulum; trc, main body cavity; vf, vacuoles with fibrillar electron-dense content; vl, vacuoles with electron-lucent content.

apical surface is entirely occupied by numerous cilia (Figs. 9A and 9B). Between these clusters, there are cells with election-gray cytoplasm. Some of them are putative secretory cells: they form small lobe-shaped apical projections which are scattered between the cilia (Figs. 9B, 9D–9F). The cell membrane of such projections is smooth and lacks microvilli and cuticle. It is worth noting that the mouth is clogged with small drop-shaped structures with very thin and short stalk (Fig. 9C). In some non-ciliated cells without apical projections, we found different stages of kinetosome biogenesis: clusters of "fibrogranular material" (Fig. 9B), deuterosome-like structures surrounded with procentrioles (Fig. 9E). Few cells in this region demonstrate mitotic figures (Fig. 9F). All cells located close to the mouth are multiciliated, and their structure is similar to that in other studied species.

## Effect of EdU concentration

Based on our experience with sponges, relatively high concentrations of EdU (100 and 300 μM) were applied to studied bryozoan species, while a suitable concentration for mammalian cell culture is 10 μM (*Borisenko et al., 2015*; *Ereskovsky et al., 2015*). The fluorescence intensity of nuclei labeled with 300 uM of EdU was much stronger. The labeling with 100 μM of EdU gave distinguishable but weak signal, and since EdU is not as toxic as BrdU, we decided to use concentration of 300 μM. We found that a three-hour incubation in EdU was not enough for "adult" polypides to incorporate the label while incubation during six hours gave a stable positive signal.

## Proliferating activity in young and adult polypides

Proliferating activity in *E. pilos*a, *A. scabra* and *R. ovata* varies depending on the age of zooids: it is the highest in developing buds and gradually decreases in zooids of the older age classes. In the young buds almost all nuclei are EdU-positive (Figs. 10A and 10B), while different parts of a given organ strongly contrast in the number of EdU-labeled cells in the late buds. Thus, the distal part of the tentacles lack positive EdU signal, although many cells still incorporate EdU within their proximal part (Figs. 10C and 10D).

In the lophophore of "young" fully developed polypides (from the 1st and 2nd age classes), we detected EdU-positive signal in the lophopore base. Several EdU-positive cells are located in the proximal and middle regions of the ciliated pits (Figs. 11A–11D), and a positive signal was detected in CP-, DP-and EP-cells. We registered from 3 up to

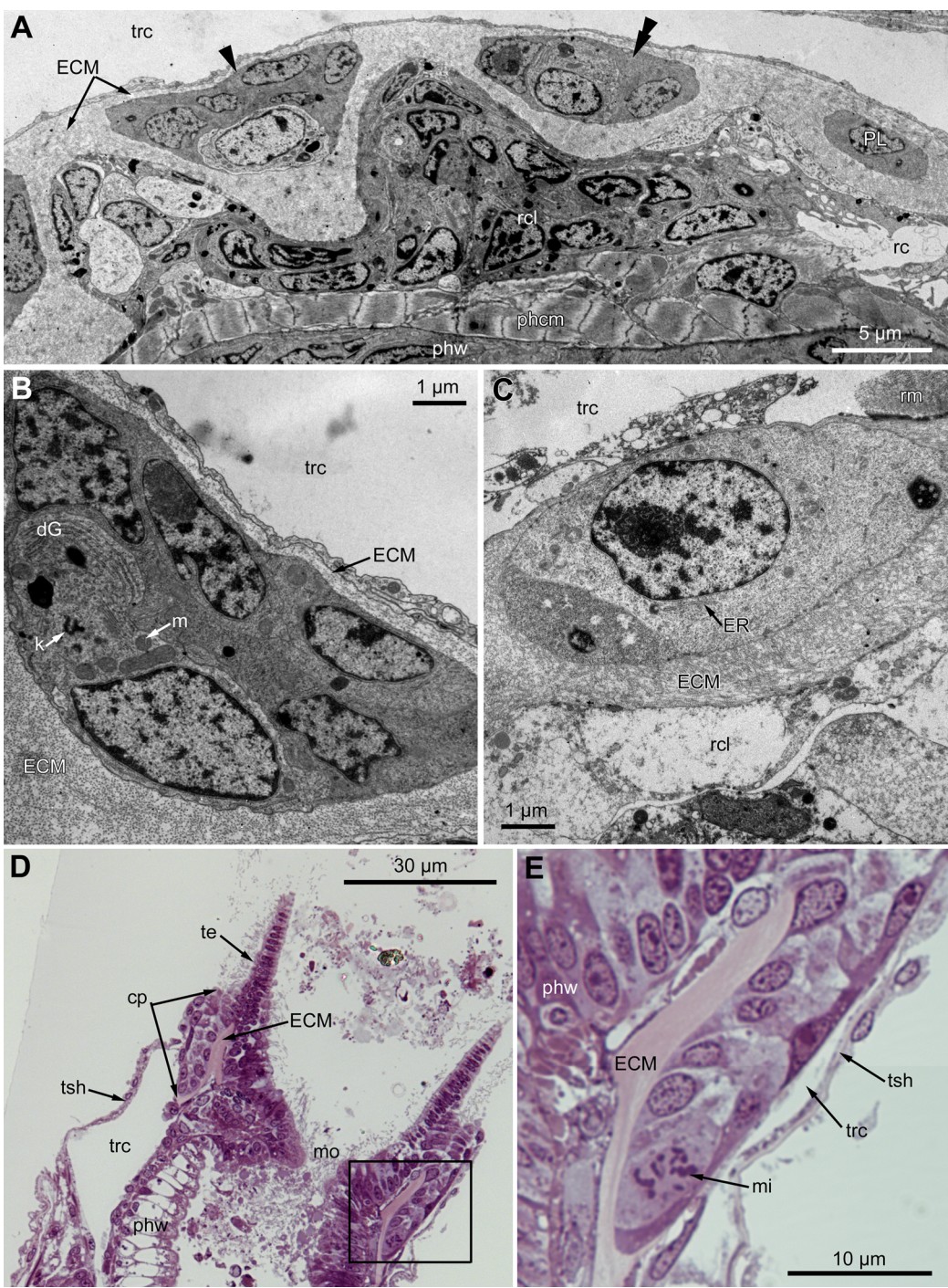

**Figure 7 Microanatomy of the proximal region of the ciliated pits in the poypides of the 3rd age class: (A and B),** *Aquiloniella scabra*; **(C–E),** *Rhamhostomella ovata*. (A) Oblique-cross sections of the lophopore base (TEM; mouth downwards), showing three ciliated pits cut at different levels: close to the border with middle region (arrowhead), close to the bottom (double arrowhead), through the most basal P-cell (PL). (B) Ultrastructure of P-cells located close to the border with the middle region (oblique-cross sections of the lophopore base; TEM; mouth downwards). (C) Ultrastructure of P-cells at the bottom of the ciliated pit (oblique cross-sections of the lophopore base; TEM; mouth downwards). (D) Longitudinal semi-thin section of the polypide through the wall of the ciliated pit (tentacle tips upwards). The white frame corresponds to the enlarged area in (E). (E) Enlargement of (D),

**Figure 7** (continued)
demonstrating mitotic figure within one of P-cells. Abbreviations: dG, dictyosome of Golgi complex;
ECM, extracellular matrix; ER, endoplasmic reticulum; k, kinetosome; mi, mitotic figure; mo, mouth;
phcm, circular muscles of the pharynx (pharynx constrictors); phw, epithelium of the pharynx; PL, the
lowest P-cell at the bottom of ciliated pit; rc, ring coelomic canal of the lophophore; rcl, coelomic lining of
the ring canal; te, tentacle; trc, main body cavity; tsh, tentacle sheath.

16 EdU-positive cells per ciliated pit. This parameter varies across ciliated pits and across polypides of the same age class for each studied species (Table 1). The number of EdU-positive cells per ciliated pit was less in *E. pilosa* than that in *R. ovata* and *A. scabra* (Table 1). Close to the outer rim of the oral region, EdU-positive cells are arranged in clusters at the level of the distal and middle parts of the ciliated pits (Figs. 11B and 11E). At the abfrontal tentacle surface, the EdU label was incorporated by individual E-cells at the proximal end of the tentacles (Figs. 11A and 11D). In addition, in four young polypides of *E. pilosa*, individual EdU-positive E-cells were also scattered within the proximal part of the tentacle (Fig. 11F). In several polypides of *E. pilos*a, EdU-positive nuclei were also detected in the walls of the intertentacular organ during its formation.

In all studied species, "adult" polypides of the 4th and 5th age classes demonstrate EdU-positive signal only in the lophophore base (except for the polypides of *E. pilosa* during the formation of the intertentacular organ). Though our study is qualitative and we did not statistically analyze the proliferating activity at the lophophore base, the number of EdU-positive cells per the ciliated pit in "adult" polypides is less than in "young" ones (Table 1). A larger number of EdU-positive cells in "young" polypides probably indicates that their tentacles are still lengthening until their lophophores acquire a specific shape.

## Proliferating activity during tentacle regeneration

We traced changes in proliferating activity within the lophophores of adult polypides (5th and 6th age classes) of *E. pilosa* during tentacle(s) regeneration (Fig. 12). A total of 24 h after surgery, we detected a single (or rarely two) EdU-positive cell at the abfrontal tentacle surface (E-cell) near wound healing. Later on, 48 h after surgery, in addition to these labeled E-cells, an EdU-positive signal is detected in the cells of the tentacle coelomic lining, namely in the epiperitoneal cells (Figs. 12A–12D). Labeled cells of the coelomic lining were located at a distance of about 50–70 μm from wound healing, as well as in the proximal part of the tentacle (Figs. 12C and 12D). EdU-positive cells were absent in the distal part of the intact tentacles. The labeling pattern at the lophophore base is similar to the control zooids, although the number of EdU-positive cells was greater at the base of the wounded tentacles. After 3 days (72 h after surgery), the label is still present near the wound: both in the E-cells and in the cells of coelomic lining. Up to 120 h after surgery, the distribution of EdU-labeled cells remains the same (Figs. 12E–12G). By the 7th day of regeneration (168 h after surgery), the proliferating activity in the regenerating lophophores is similar to the control ones, and the E-cells near the wound do not incorporate EdU anymore. Interestingly, in one of the polypides, the label was detected in the cells of the coelomic lining at the base of the wounded tentacles. The level of the

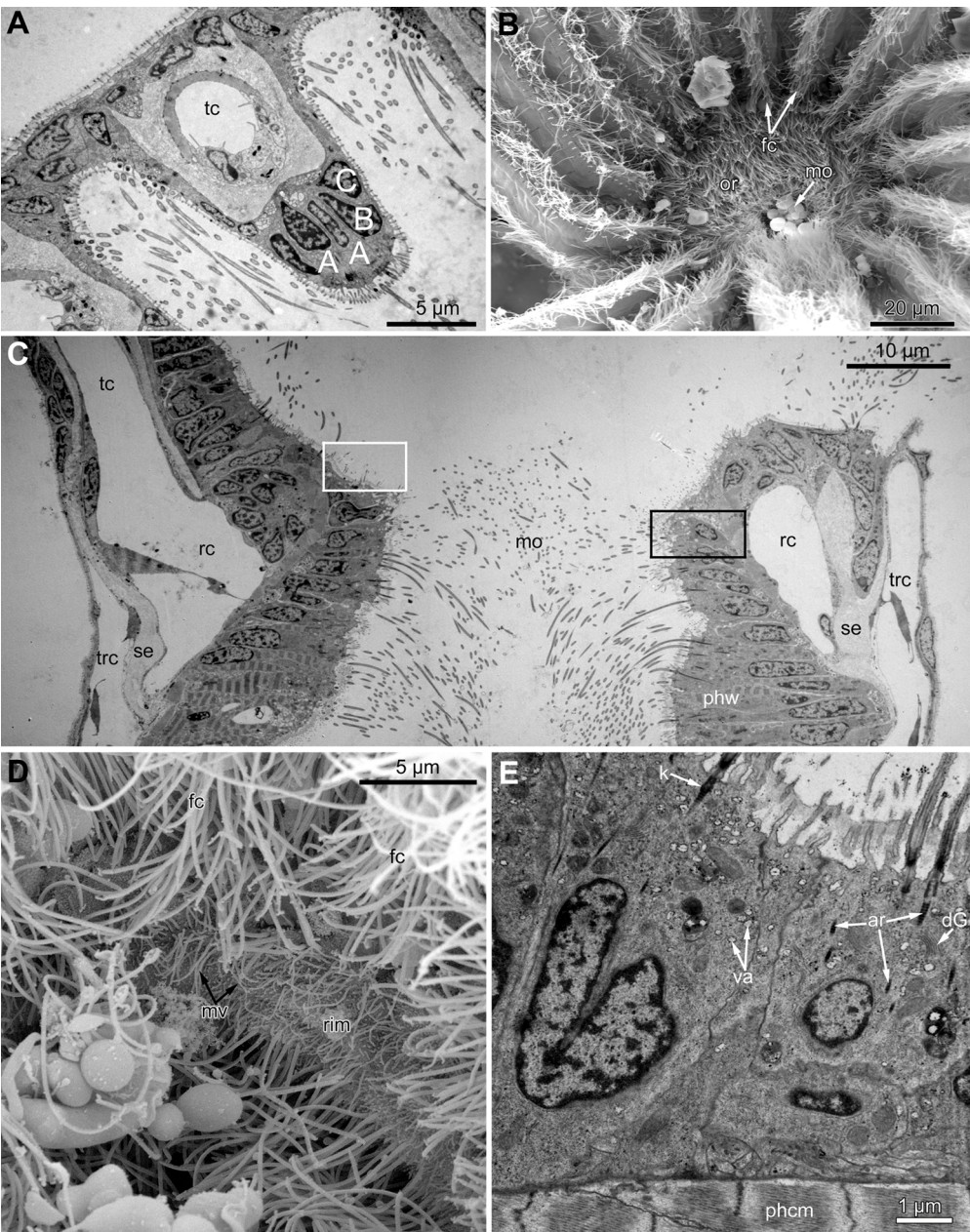

**Figure 8 Microanatomy of the oral region: (A–C and E)** *Aquiloniella scabra*; **(C and D)** *Eucratea loricata.* (A) The frontal row of A-cells is two cells wide at the very base of the tentacles (cross section; TEM). (B) The tentacle bases and the oral region of the polypide (SEM). (C) Longitudinal section (TEM) of the polypide through the tentacle (left) and the ciliated pit (right). The white frame approximately corresponds to the outer rim shown in (D). The black frame approximately corresponds to the ciliated area of the oral region shown in (E). (D) The outer rim of the oral region lacks cilia and possesses long microvilli in *E. loricata* (SEM). (E) The fine structure of the ciliated cells of the oral region (oblique-cross section; TEM). Abbreviations: A, frontal cell (A-cell); B, latero-frontal cell (B-cell); C, fronto-lateral cell (C-cell); ar, axial rootlet; dG, dictyosome of Golgi complex; fc, frontal ciliary band; k, kinetosome; mo, mouth; mv, microvilli; or, oral region; phcm, circular muscles of the pharynx (pharynx constrictors); phw, epithelium of the pharynx; rc, ring coelomic canal of the lophophore; rim, outer rim of the oral region; se, septum dividing the lophophore and main body cavity; tc, tentacle coelom; trc, main body cavity; va, vacuole.

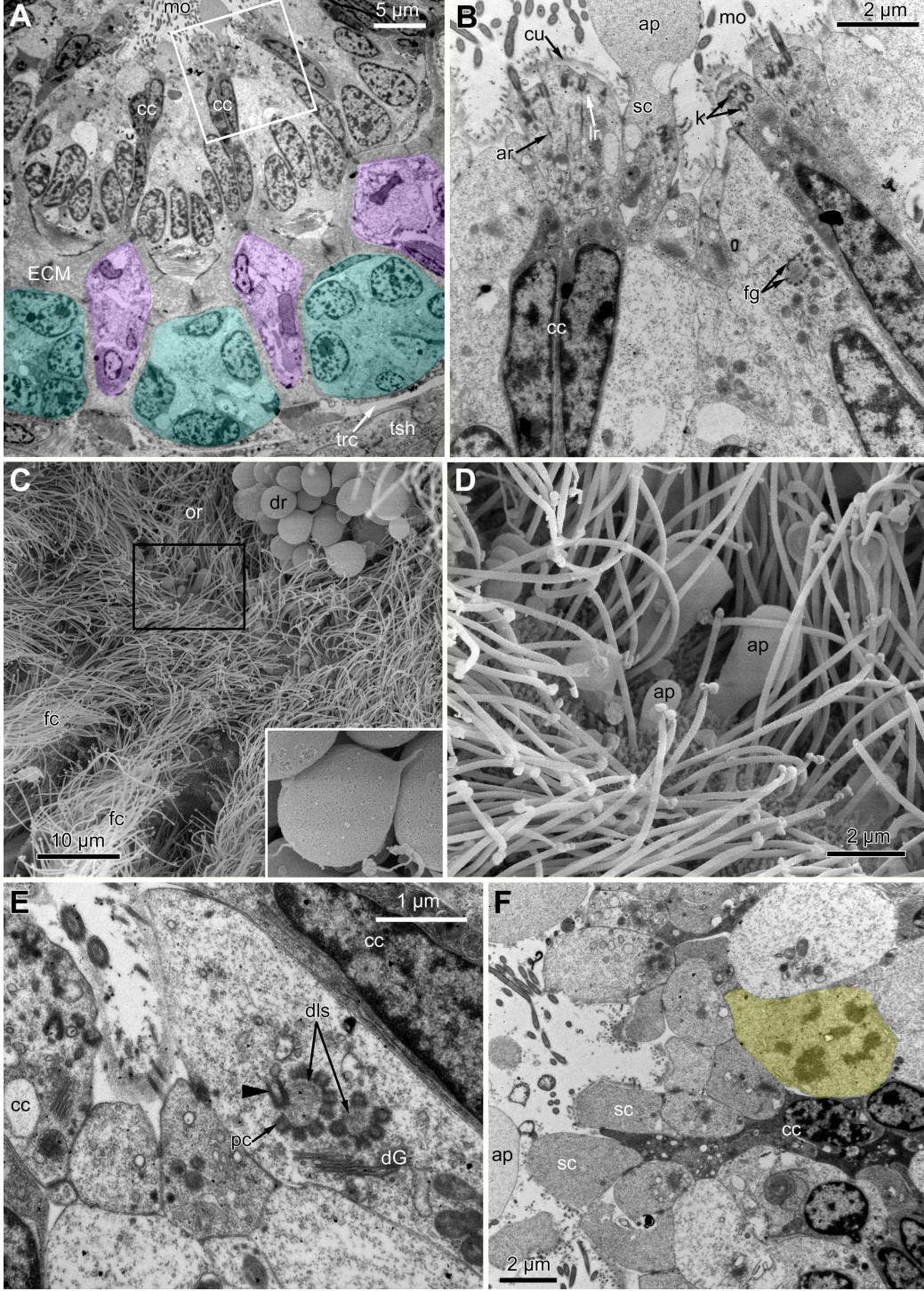

**Figure 9 The fine structure of the oral region in the polypides of the 3rd and 4th age classes in *Rhamphostomella ovata*.** (A, B, E and F) Cross sections of the lophophore base (TEM); (C and D) the oral region of polypide (SEM). (A) The oral region is composed of two cell types: ciliated cells with electron-dense cytoplasm (located in the continuation of the rows of A-cells) and non-ciliated cells with electron-gray cytoplasm, forming small lobe-shaped apical projections. The tentacle coeloms are shaded in violet, the cells composing ciliated pits are shaded in cyan. The white frame corresponds to the enlarged area in (B). (B) Enlargement of (A), showing the fine structure of the ciliated cells of the oral

**Figure 9** (continued)
region. Note cluster of "fibrogranular materiall" in one of the non-ciliated cells. (C) General view of the oral region of polypide. Note the drop-shaped structures clogging the mouth. The black frame corresponds to enlarged area in (D). Inset: Drop-shaped structures have thin and short stalk. (D) Enlargement of (C), showing apical projections of putative secretory cells. (E) Details of kinetosome biogenesis in non-ciliated cells. Note that one of the procentrioles (arrowhead) is larger and lies at the right angle to the rest. (F) Cell undergoing mitosis (shaded in yellow). Abbreviations: ap, apical projection of putative secretory cell; ar, axial rootlet; cc, ciliated cells of the oral region; cu, cuticle; dG, dictyosome of Golgi complex; dls, deuterosome-like structures; dr, drop-shaped structures clogging the mouth; fc, frontal ciliary band; fg, "fibrogranular material", k, kinetosome; lr, lateral rootlet; mo, mouth; or, oral region; pc, procentrioles; sc, putative secretory cell; trc, main body cavity; tsh, tentacle sheath.

EdU incorporation in these nuclei is low compared with those of the epidermis of the lophophore base or the epidermis of the intratentacular organ, but is clear distinguishable from the nonspecific background.

The wounded tentacles did not restore their original length or even considerably lengthen during the experiment (7 days).

## DISCUSSION

### Ciliated pits

In the literature, there are only two brief mentions of ciliated (or intertentacular) pits of unclear function in Gymnolaemata: in *Cryptosula pallasiana* (*Gordon, 1974*) and in *Hislopia malayensis* (*Schwaha & Wood, 2011*). Both studies provide only a general overview, and we are apparently the first to provide a detailed description of the ultrastructure of the ciliated pits in cheilostomes and suggest their possible function (see below). According to our data, the proximal and middle parts of the ciliated pits contain stem cells undergoing mitosis (Figs. 3F and 7E), and the cells within the middle part demonstrate different stages of kinetosome biogenesis (Fig. 6). Though Gordon did not mention this in the text, one of his illustrations (*Gordon, 1974*, fig. 6) demonstrates the presence of irregularly arranged kinetosomes in the apical part of one cell and the mitotic figure in another. Needless to say, that our data are in a good agreement with these findings. Based on this and our data on proliferating activity within the lophophore (see below), we can regard the ciliated pits as a specific region providing a tentacle elongation (growth) in polypides.

*Schwaha & Wood (2011)* reported that the ciliated pits in *H. malayensis* are twice as deep as in *C. pallasiana*. We also detected high variability of this parameter in the studied species. It is possible that the depth of ciliated pits reflects the differences in the structure of the lophophore base.

### Proliferating activity within the lophophore

The proliferating activity within the bryozoan lophophore has never been described before. However, the presence of blastemic cells has been mentioned twice: in the oral region of a polypide in *C. pallasiana* (*Gordon, 1974*) and close to the ganglion of a degenerating female polypide in *Alcyonidium polyoum* (*Matricon, 1963*). Our data revealed the presence of stem cells in the ciliated pits and in the oral region of a polypide but never in the

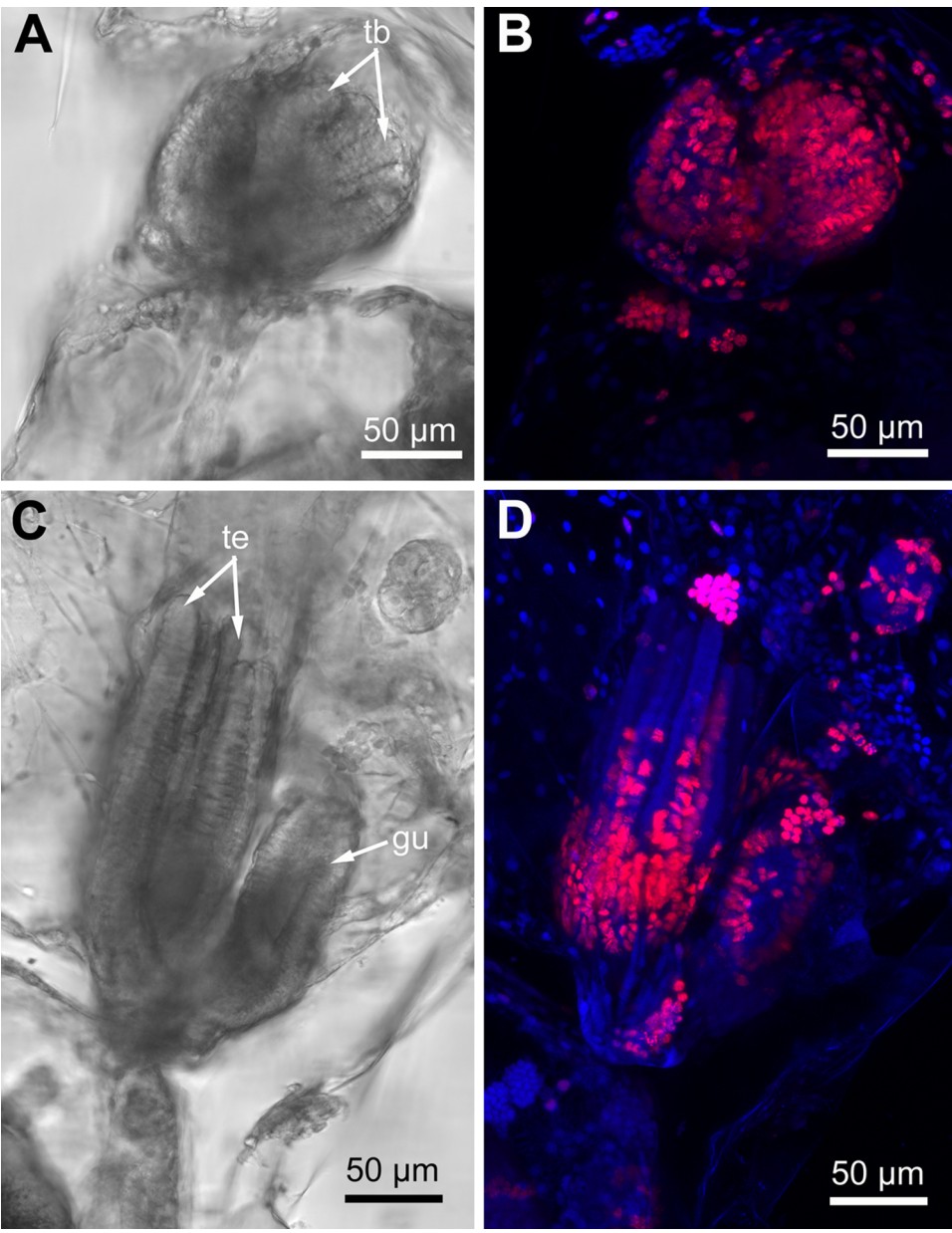

**Figure 10 Proliferating activity in buds of *Rhamphostomella ovata* (Red—EdU-labeling; blue—Hoechst).** (A) General view of the young bud (light microscopy). (B) EdU-labeling of the young bud (CLSM, the whole mount, maximal projection). (C) General view of the late bud (light microscopy). (D) EdU-labeling of the late bud (CLSM, the whole mount, maximal projection). Abbreviations: gu, gut; tb, tentacle buds; te, tentacles.

tentacles of adult polypides (Figs. 3F, 7E, 9E and 11). In addition to this, we also found different stages of kinetosome biogenesis in prospective multiciliated cells undergoing specialization (Figs. 6, 9B and 9E). Interestingly, even in the late buds, the EdU-label is restricted to the proximal part of the tentacle and was never registered in the distal part or close to the tentacle tip. These features, together with the absence of proliferating cells in the tentacles of adult polypides, demonstrate the early specialization of ciliated cells

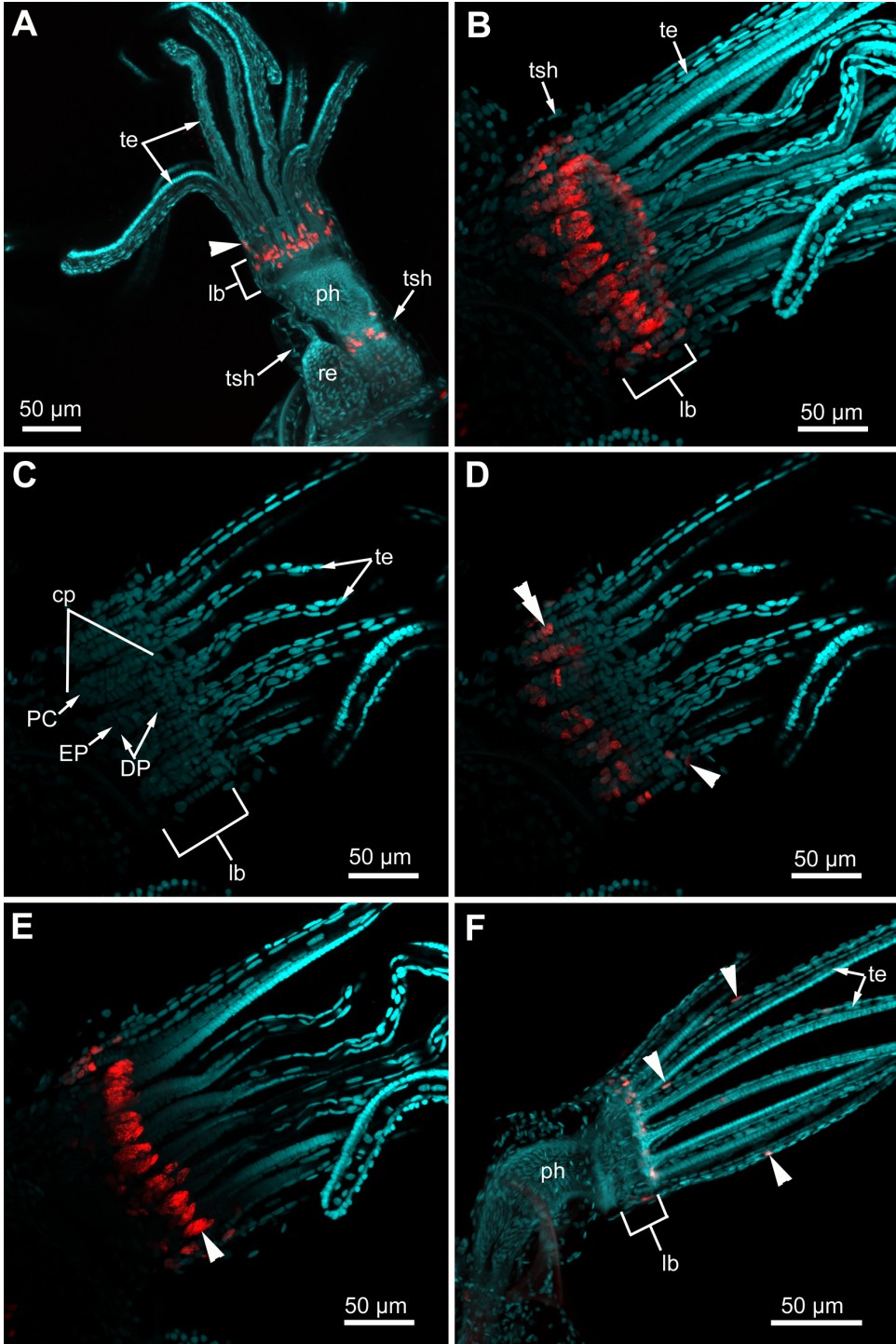

**Figure 11 Proliferating activity in "young" polypides (CLSM; maximal projections; Red—EdU-labeling, cyan—Hoechst).** (A) The lophophore of the young polypide (1st age class) in *Aquiloniella scabra* showing EdU-labeled cells (arrowheads) in the lophophore base and at the abfrontal surface of the tentacle base (whole mount). (B) The lophopore base of young polypide (1st age class) in *Rhamphostomella ovata* demonstrating EdU-labeled cells in the lophophore base (whole mount). (C) The lophophore base of the same polypide in *R. ovata*; a substack of 11 slices (from seven to 17) showing location of DP-, EP-and P-cells within ciliated pits. (D) The lophophore base of the same polypide in *R. ovata*; a substack of 11 slices (from seven to 17) demonstrating EdU-positive cells at the abfrontal surface of tentacle

**Figure 11 (continued)**
bases (arrowhead) and within the proximal and middle regions of ciliated pits (double arrowhead). (E) The lophophore base of the same polypide in *R. ovata*; a substack of 12 slices (from 27 to 38) demonstrating clusters of EdU-positive cells within the oral region (arrowhead). (F) The lophophore of young polypide (1st age class) in *Electra pilosa* showing individual EdU-positive cells scattered along the abfrontal surface in the proximal part of tentacles. Abbreviations: CP and DP, vertical rows of the cells composing the lateral walls of the ciliated pit and located below the corresponding C-and D-cells of the neighboring tentacles; EP, cells composing the abfrontal surface of the ciliated pit; PC, P-cells (cells located in the proximal region of the ciliated pit); cp, ciliated pit; lb, lophophore base; ph, pharynx; re, rectum; te, tentacles; tsh, tentacles sheath.

**Table 1 The average number of EdU-positive sells per ciliated pit in studied species.**

| Species | *Rhamphostomella ovata* | | *Aquiloniella scabra* | | *Electra pilosa* | |
|---|---|---|---|---|---|---|
| Polypide age | Young | Adult | Young | Adult | Young | Adult |
| Number of studied polypides | 12 | 9 | 11 | 8 | 10 | 14 |
| Number of studied ciliated pits | 109 | 99 | 68 | 67 | 79 | 134 |
| Average number of EdU-positive cells per ciliated pit (±SE) | 13.7 ± 0.21 | 8.7 ± 0.18 | 5.2 ± 0.15 | 3.4 ± 0.07 | 3.7 ± 0.17 | 1.8 ± 0.09 |

**Note:**
"Young" polypides belong to the 1st and 2nd age classes, "adult" polypides belong to 4th and older age classes. SE, standard error.

and provide evidence of intercalary growth of tentacles. Surprisingly, in phoronids "cells undergoing mitosis are often seen" within the tentacle epidermis, "suggesting a proliferation of epidermal cells for tentacular growth and for substitution of dead or damaged cells" (*Pardos et al., 1991*, p. 81 for both quotations).

According to our data, proliferating cells in the ciliated pits are the source for cells that compose the lateral tentacle surfaces (C-and D-cells), while the cells of the frontal tentacle surface (A-and B-cells) originate from proliferating cells of the oral region. It is very likely that CP-and DP-cells give rise to C-and D-cells, respectively, since we traced the stages of kinetosome biogenesis in these multiciliated cells. However, we are not so positive about the source for E-and F-cells. We detected an EdU-positive signal in EP-cells and in the most basal E-cells but we did not trace the fate of these labeled cells. During the tentacle growth/elongation, F-cells are probably formed "in advance" compared to E-cells, since we registered clusters of tightly packed F-cells in the most basal part of the tentacle in young polypides (Fig. 2C). We did not detected specific source for the cells of tentacle coelomic lining. It is probably located near the proximal end of the tentacles.

This study revealed the possible function of the ciliated pits: it is very likely that they participate in the tentacle elongation in young and adult polypides. The specific location of the proliferating sites within the lophophore base provides a possibility of tentacle elongation without interfering with the feeding process. Our results are in a good agreement with observations made by *Dick (1987)* who described a transformation of obliquely truncated lophophore to equitentacled one, and vice versa, for two cheilostomes *Holoporella brunnea* (=*Celleporaria brunnea*) and *Membranipora serrilamella* (=*M. villosa*). Polypides with obliquely truncated lophophores usually are located at the colony periphery and also surround the chimneys (excurrent water outlets

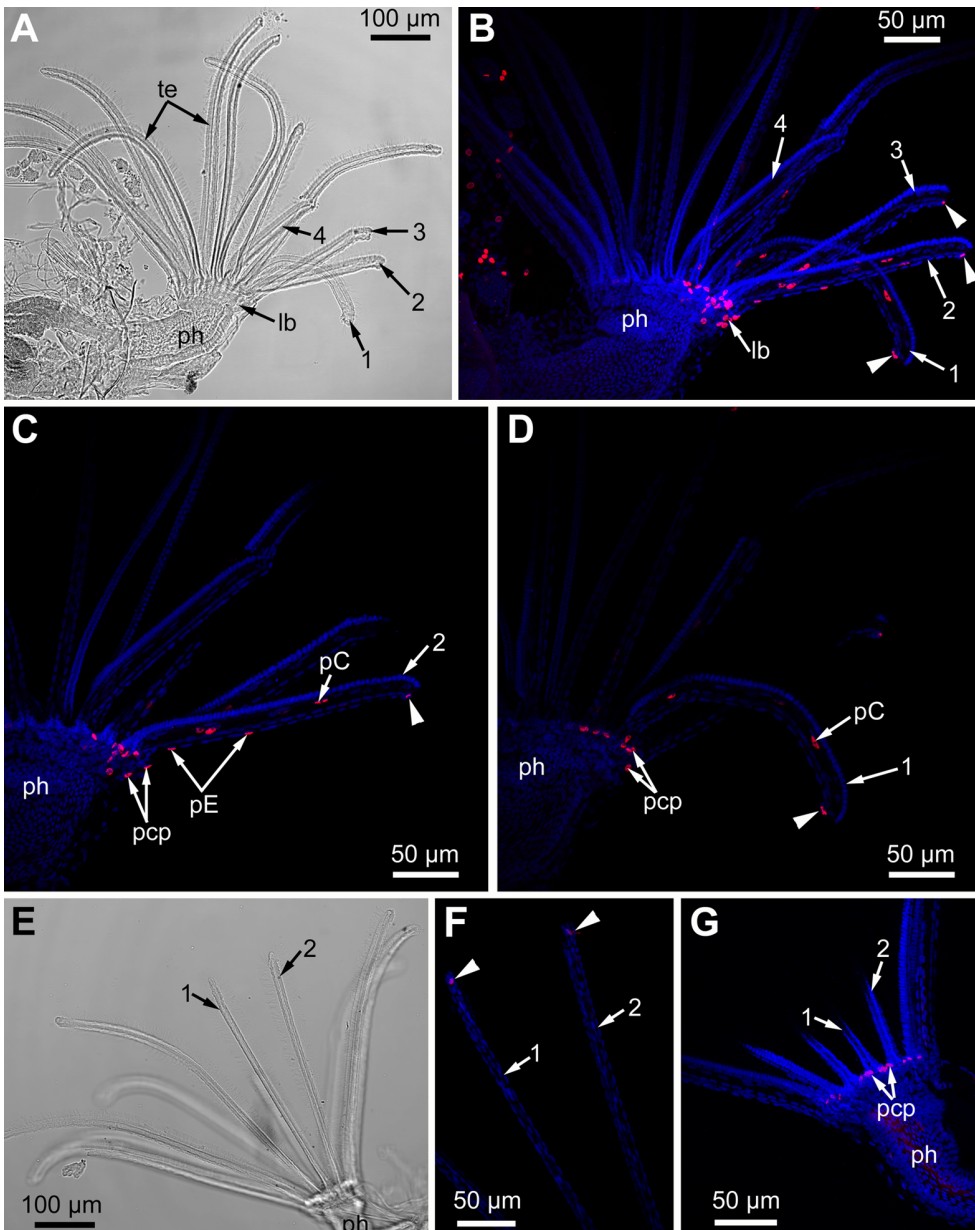

**Figure 12 Proliferating activity in the lophophore of adult polypides (6th age class) in *Electra pilosa* during experimental tentacle regeneration.** Red—EdU-labeling, blue—Hoechst; (A–D), 48 h after operation; (E–G), 120 h after operation. (A) General view of the polypide with regenerating tentacles (light microscopy); 1-4—numbered wounded tentacles. (B) Distribution of EdU-positive cells in the lophophore of the same polypide (CLSM, whole mount). Note a EdU-positive signal in E-cell (arrowheads) close to the wound. (C) Distribution of EdU-positive cells in the operated tentacle #2 of the same polypide (CLSM, substack of 13 slices: from seven to 19). Note a EdU-positive signal in E-cell (arrowhead) close to the wound. (D) Distribution of EdU-positive cells in the operated tentacle #1 of the same polypide (CLSM, substack of 14 slices: from 32 to 45). Note a EdU-positive signal in E-cell (arrowhead) close to the wound healing. (E) General view of the polypide with regenerating tentacles (light microscopy). (F) The tips of regenerating tentacles of the same polypide (CLSM, substack of 28 slices: from 8 to 35). Note a EdU-positive signal in the E-cell (arrowheads) close to the wound. (G) The lophophore base of the same polypide showing EdU-positive cells in the ciliated pits (CLSM, maximal projection of half a polypide).

within encrusting colonies—e.g., see *Banta, McKinney & Zimmer, 1974*; *Cook, 1977*; *Winston, 1978*, *1979*; *Cook & Chimonides, 1980*; *Dick, 1987*). Elongated tentacles of obliquely truncated lophophores fringe the chimney area. It was demonstrated that the location of the chimneys within a given colony changes with time (*Von Dassow, 2005a*, *2005b*, *2006*) and an alteration of colony-wide water currents requires a remodeling of the lophophore shape in the polypides surrounding the newly formed chimney. Our results provide evidence for a potential altering of the lophophore shape in polypides, which is crucial for encrusting colonies that need to rearrange colony-wide water currents with time.

## Kinetosome biogenesis

The kinetosome biogenesis has been studied in great detail (both morphological and molecular aspects) in different vertebrates, while similar data for invertebrates, protists and plants are rather scarce. Nevertheless, several general pathways are recognized: canonical (or centriolar) and non-canonical (or acentriolar, including deuterosome-mediated and several de novo mechanisms)—for a review, see *Nabais, Pereira & Bettencourt-Dias (2017)*. Deuterosome-mediated assembly of kinetosomes was first described for foetal lung in rat (*Sorokin, 1968*) and now it has been proved to be the main pathway in different multiciliated epithelia in vertebrates (*Dirksen, 1991*; *Hagiwara, Ohwada & Takata, 2004*; *Al Jord et al., 2014*; *Balestra & Gönczy, 2014*; *Yan, Zhao & Zhu, 2016*; *Spassky & Meunier, 2017*; *Shahid & Singh, 2018*). It was suggested that the deuterosome-mediated pathway is specific for vertebrates (*Zhao et al., 2013*; *Nabais, Pereira & Bettencourt-Dias, 2017*). It should be mentioned that for invertebrates, the morphological comparison by Nabais and coauthors (*Nabais, Pereira & Bettencourt-Dias, 2017*) was based solely on data obtained during multiciliated spermiogenesis in mollusks and insects.

As for multiciliated epithelial cells in invertebrates, the kinetosome formation has been described only for a few species. Among them, the centriolar pathway, as a single mode for kinetosome producing, was reported for prototroch cells in trochophore of *Nereis limbata* Ehlers, 1868 (=*Alitta succinea* (Leuckart, 1847)) (*Kalnins, 1967*). A combination of the centriolar pathway and the de novo mechanism was reported for embryos of turbellarian *Macrostomum hystricinum* (*Tyler, 1981*) and acoel *Archaphanostoma* sp. (*Tyler, 1984*). It was demonstrated that kinetosomes are formed solely by the de novo mechanism in the gill filaments of scallops (*Ash & Stephens, 1975*), the body wall of adult turbellarians (*Cifrian, García-Corrales & Martínez-Alos, 1992*; *Drobysheva, 2006*, *2010*), and macrocilia in ctenophores (*Tamm & Tamm, 1988*). In all described cases of the acentriolar pathway in invertebrates, specific clusters of fibrous granules were present, but no structures resembling deuterosomes have ever been described. In addition to these fibrous granules (or "fibro-granular material") in bryozoans, we also found that procentrioles are formed around electrone-dense spherical bodies located close to the dictyosomes of Golgi complex in the apical part of the cells (Figs. 6C and 6E). These spherical bodies are similar to deuterosomes described for vertebrates (*Sorokin, 1968*; *Anderson & Brenner, 1971*; *Dirksen, 1971*, *1991*; *Hagiwara et al., 2000*; *Hagiwara, Ohwada & Takata, 2004*). Unfortunately, any data about molecular aspects of

kinetosome biogenesis in invertebrates are absent. This fact and also the lack of the information about kinetosome biogenesis in many groups of invertebrates, prevent us from any further comparisons and speculations.

### Tentacles regeneration

This study demonstrated that microsurgical amputation of distal part of tentacle(s) results in the reparation of damaged tentacles and does not cause an extra cycle of degeneration-regeneration of the whole polypide. It is interesting that only E-cells (abfrontal non-ciliated cells) located at the wound edge are responsible for its healing. Labeling with EdU demonstrates that the wound causes their dedifferentiation and limited proliferation from 1st to 5th day after surgery (Fig. 12) to produce cells providing wound closure. In intact tentacles (in both experimental and control polypides), epidermal cells never incorporate EdU. The ciliated cells of the tentacles are very specialized and unable to proliferate even under the necessity of wound healing. Surprisingly, regenerating tentacles in polypides of *E. pilosa* did not even considerably lengthened by the 7th day after surgery. This contrasts with the data obtained from plylactolaemates (*Otto, 1921*; *Oda, 1954*) whose tentacles restore their length in about 10 days. We suggest that the tentacle recover its length according to a mechanism similar to normal growth, powered by cell proliferation both in the ciliated pits and the oral region. We did not statistically analyze proliferating activity at the lophophore base since our study is qualitative and this is the first attempt to test the ability of tentacle regeneration in marine bryozoans.

Among the different variants of regeneration processes in animals, two main types are recognized: epimorphosis and morphallaxis (*Agata, Saito & Nakajima, 2007*). During epimorphosis, the cells of the residual part of the organ dedifferentiate and form a temporary structure—the blastema. The dedifferentiated cells proliferate in the blastema and produce material for de novo formation of the lost part of the organ. Alternatively, no blastema is formed during morphollaxis, and regeneration of the lost body part is accomplished through remodeling of the remaining part of the body. In this case, there is no accumulation of undifferentiated cells or specific site of massive proliferation. According to our data, the regeneration of tentacles in *E. pilosa* is similar to the morphollaxis type. Cellular mechanisms, de- and transdifferentiation events remain to be described in the future studies.

## CONCLUSIONS

This study demonstrates that the specific design of the bryozoan lophophore base enable intercalary growth of tentacles. Using TEM and EdU-labeling, we detected the presence of stem cells within the ciliated pits and the oral region of polypides and never within tentacles of adult polypides. Cells of the ciliated pits are the source for the cells composing the lateral tentacle surfaces, while the cells of the frontal tentacle surface originate from proliferating cells of the oral region. Thus, the ciliated pits and the oral region of polypide are specific sites responsible for the tentacle elongation. This confirmed the

hypothesis that cell proliferation during tentacle elongation takes place at the lophophore base and hence provides an opportunity of the tentacle elongation in adult polypides without interfering with the feeding process. Such a potential to alter the lophophore shape is crucial for encrusting colonies when they require the rearrangement of colony-wide water currents with time. For the first time in prospective multiciliated epithelial cells in invertebrates, we registered deuterosome-like structures during the kinetosome biogenesis. Experiments on tentacle regeneration in *Electra pilosa* demonstrated that among all types of epidermal cell, only abfrontal non-ciliated E-cells are responsible for wound healing. Ciliated cells on the frontal and lateral tentacle surfaces are very specialized and unable to proliferate even under the necessity of wound healing. The regeneration of tentacles is very slow and similar to the morphollaxis type. We suggest that the tentacle recovers its length by a mechanism similar to normal growth, powered by proliferation of cells both in the ciliated pits and the oral region.

## ACKNOWLEDGEMENTS

We sincerely thank Sergey Bagrov for help with sampling and Dr. Andrej Dobrovolskij for inspiring discussion and valuable comments (both—St. Petersburg State University). Gratitude is due to the Research and Educational Station "Belomorskaia" (former Marine Biological Station) of St. Petersburg University for hospitality. The study was partly done at the Research center "Molecular and Cell Technologies" (St. Petersburg State University).

### Funding

This study was supported by St. Petersburg State University (projects 0.40.485.2017 and 1.42.1099.2016). The funders had no role in study design, data collection and analysis, decision to publish, or preparation of the manuscript.

### Grant Disclosures

The following grant information was disclosed by the authors:
St. Petersburg State University: 0.40.485.2017 and 1.42.1099.2016.

### Competing Interests

The authors declare that they have no competing interests.

### Author Contributions

- Natalia Shunatova conceived and designed the experiments, performed the experiments, analyzed the data, prepared figures and/or tables, authored or reviewed drafts of the paper, and approved the final draft.
- Ilya Borisenko performed the experiments, analyzed the data, prepared figures and/or tables, authored or reviewed drafts of the paper, and approved the final draft.

## Data Availability

All original images are available at Figshare: Shunatova, Natalia (2020): A public data set for "Proliferating activity in a bryozoan lophophore". figshare. Dataset. DOI 10.6084/m9.figshare.12010992.v1.

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
