# Peer review of "Proliferating activity in a bryozoan lophophore"

_PeerJ, doi:10.7717/peerj.9179_

## Round 0.1 · original submission · Major Revisions

Dear Dr. Shunatova,

Thank you for submitting your manuscript to PeerJ. Your manuscript has been evaluated by three peer reviewers, and their comments are appended below.

The reviewers think the topic is very interesting and represent a significant contribution to our understanding of the lophophore and animal cell structure which have been poorly studied in bryozoans. However, reviewers consider the manuscript needs some changes in the different sections (e.g. structure, terminology, interpretation of the data).

Based on the referees' recommendations, I arrive at this decision: The manuscript does merit publication in PeerJ but it is not acceptable in its current form and needs a major revision based on the reviewer's comments below. I therefore invite you to resubmit a revised version. Please carefully consider the comments of the reviewers and provide a point-by-point response which clearly defines the changes made. Note that reviewer 1 has also provided his/her advice in an annotated pdf file.

Please also have the language in your manuscript checked by a native English-speaking colleague as reviewers 1 and 3 consider the manuscript needs a through revision for correcting typos. Also, PeerJ can provide language editing services - please contact us at editorial.support@peerj.com for pricing (be sure to provide your manuscript number and title).

Thank you for your patience with the evaluation process and for choosing PeerJ.

I look forward to receiving your revised manuscript.

Yours sincerely,


Blanca Figuerola
* * *
Academic editor
PeerJ
* * *
Reviewer 1 ·

Basic reporting

In their article entitled "Proliferating activity in a bryozoan lophophore: a case study in gymnolaemates“ Shunatova and Borisenko describe the occurence of proliferating cells in the lophophores of three species of gymnolaemate bryozoans. Proliferating cells were visualised by incorporation of labeled nucleotide analogues and imaged using confocal microscopy. In addition the fine structure of the lophophores and the tentacle bases was described in the same plus one additional species. This investigation is very interesting, because patterns of cell division during growth and organ formation in bryozoans have so far not been studied. The results can help explaining how different lophophore shapes arise within a colony and how regeneration happens.

The language needs much improvement. The text is full of errors and typos. The most common mistakes concern the use of articles and prepositions, tense, as well as singular and plural forms. I have examplary made corrections in the abstract and on the first page of the Introduction secions, but can not fulfil this for the whole manuscript.

The topic is introduced and set into context of the existing literature. Some gaps concern the references of stem cells and proliferation in other animal groups (indicated in the pdf). Also, the use of the term "stem-like cell“ is not clear to me. I have so far only seen this term used for cancer stem cells. If you use this term here you should very clearly define what it means, and how it differs from "stem cell“. Otherwise I would recommend using the latter.

The figures provided are of high quality. They illustrate very well the findings and complement the descriptions given in the results section. Figure 4 is a bit difficult to understand, because the coloration for vertical and horizontal planes don’t make sense in the perspective drawing. I would recommend here to simply remove the lower green part, or to put two drawings side by side, one showing a horizontal plane, the other a vertical plane

Experimental design

The methods chosen are appropriate and described in sufficient details. I can see no major flaws.

Validity of the findings

The descriptions of the ultrastructure are very detailed and thoroughly discusse. the finding of a deuterosome is indeed interesting and worth to be reported, even if comparative data are scarce.

One point that remains a bit more elusive is the fate of the proliferating cells, both during normal growth as well as regeneration. To clearly show cell fates and the potency of the supposed stem cells, more EdU experiments with pulse-chase treatments are neccessary. However, these can be extensive, so the authors might want to consider this for a follow-up study. For this study I would simply recommend wording their interpretations in lines 399-409 a bit more careful (e.g. avoid "absolutely sure“ and such expressions.

Annotated reviews are not available for download in order to protect the identity of reviewers who chose to remain anonymous.

·

Basic reporting

English is good
Literature references are mostly sufficient
Row data of confocal sudy are NOT shared (!)
Interesting hypotheses and results

Experimental design

The primary research is original
experimental design is well done

Validity of the findings

no comment

Additional comments

The study is devoted to comprehensive investigation of the tentacle apparatus of four bryozoans species. The study is done with the use of all modern zoological methods, which allow to obtain very interesting new results. This study contains perfect photographs and many schemes. The study may be published in “PeerJ” after correction, accordingly to the list below.
Major comments:
1) In current version, the Introduction consists of several parts, which seem to be disconnected with each other. It would be better to organize the Introduction in logical way.
I recommend authors to start the Introduction from the description of bryozoans, their anatomy, phylogeny, and some peculiarities of life style and feeding. Then, mention why the proliferating activity is important for these animals and how the proliferating activity is studied at this moment concerning bryozoans and other animals. After all, to formulate the problem of the study.
2) In Material and methods, define the chapter “Animals” (or “Collection of animals”) where provide all information concerning name of species, place of their collection, methods f collections, ect.
3) Because most of results are devoted to organization of the tentacles and lophophore of several bryozoan species, the title of the paper needs some correction. Nine of twelve figures of the manuscript are devoted to the description of the organization of the lophophore and tentacles! To my mind, the information about organization of the lophophore must be incorporated into title of the manuscript.
It is difficult to recommend something and authors know better how to change the title…. I can recommend following variants: “Organization and proliferating activity of the lophophore….” “Fine structure and….”, “Details of morphology and ….”.
4) I recommend authors to prepare scheme or (better) 3D reconstruction of the lophophore base, in which the tentacles and ciliated pits must be shown. It is important for better understanding of morphology of the lophophore base. Thus, this scheme may allow reader to understand the location of the ciliated pits in respect to the tentacles and the mouth.
5) Some important recommendations for preparation of figures must be accepted

Minor comments:
P.7, L. 59 “gut and associated muscles and nervous system” must be changed to “gut and associated muscles, and nervous system”
P.7, L. 74 “….and, correspondingly, and….” must be changed
P. 8, L. 110. “At the bases tentacles unite and…” replace to “At the bases, tentacles unite each other and….”
P. 8, L. 119-120. Please provide the references for phrase “It is well known that different benthic animals predate on bryozoans and usually they consume a whole polypide or considerable part of it.”
P. 9, L. 144. “Aquiloniella scabra van Beneden, 1848 and..” replace to “Aquiloniella scabra van Beneden, 1848, and…”
P. 9, L. 146. “E. pilosa and A. scabra…” replace to “E. pilosa, and A. scabra….”
P. 9, L. 161. Please provide the name and label of the tool, with which the semithin and ultrathin sections were done.
P. 10, L. 202. “Rhamphostomella ovata and Aquiloniella scabra.” replace to “Rhamphostomella ovata, and Aquiloniella scabra.”
P. 11, L. 231. “At the very base of the tentacle…” – “very base” sounds wrong. Please rephrase. Use the term “the most proximal”
P. 12, L. 285. See my previous comment
Figures are very nice. But in some pictures, the abbreviations are not properly visible. I recommend authors to use “Stroke” tool for some abbreviations (for ex., “fc” 1D; “1µm”, “ar”, “ECM, “k” in Fig. 5; “m”, “k”, “dG” in Fig. 6 and many others).
It looks very negligently the location of the scale bar in different parts of photos within one plate. Please use the similar location of the scale bar at least in one plate.
Different size of abbreviations is used for different figures. For example the size of scale bar calibration is much bigger in Fig. 12. I recommend to use the same size for same types of abbreviations through all figures.

Reviewer 3 ·

Basic reporting

Shunatova and Borisenko have followed the PeerJ instructions to authors in formatting their paper. The authors have provided sufficient information in the introduction to provide readers with an understanding of bryozoan zooid anatomy and colony structure. The authors make good use of the existing literature to provide context for their study as it relates to bryozoans and to the broader biology of animals. The figures the authors include in the manuscript beautifully illustrate the data they describe in the results section, although some may find figure 4 difficult to interpret. The work the authors present in this manuscript is an appropriate unit of publication, although it is clear that the results presented in this submission fit into a broader research program, which will yield other publications on the structure of bryozoan polypides. Overall the paper conforms to professional standards, but there are a number of typos and grammatical mistakes, requiring that the manuscript be carefully proofread. For example, line 48: contributes should be contribute; line 49: cells should be cell; line 64: In majority should be in the majority; line 65: regenerate should be regenerates; line 73: the latter are arranged in five longitudinal rows and, correspondingly, and form five ciliary bands should be the latter are arranged in five longitudinal rows and correspond to five ciliary bands; line 131: technic should be technique; line 252: an should be the; line 272: Goldgi should be Golgi; line 298: Cell membrane should be The cell membrane; line 324: cells in per should be cells per; line 324: posistive should be positive; line 349: is higher under the operated tentacles should be was greater at the base of the wounded tentacles; lines 454 and 481: Recent should be A recent. In addition, there are many places in the text where a different word should be used. For example, line 84: Depleted should Exhalent; line 114: registered should be reported; line 119: mentioning should be report; line 229: up to three should be from two to three; line 306: Basing should be Based; lines 342, 343, 350-352, 457, 463: operation should be surgery; line 356: comparing should be compared; line 373: Basing should be Based; line 402: comparing should be compared.

Missing references – on lines 454 and 481, the authors refer to recent studies, but do not cite a reference.

Experimental design

The research the authors present in this manuscript has three parts. First, the authors investigated the organization and cell structure of the tentacles, lophophore base, and the ciliated pits in four different species of gymnolaemate bryozoans (Rhamphostomella ovata, Electra pilosa, Aquiloniella scabra and Eucratea loricata) using semi-thin sections, TEM, and SEM. Second, the authors identified regions of actively dividing cells in lophophores at different stages of development (i.e., age) in R. ovata, E. pilosa, A. scabra by using a thymidine analog (EdU) to label replicating DNA which the authors imaged using a confocal microscope after chemically attaching the fluorochrome Cy5 to the incorporated analog. Lastly, the authors used their method of identify dividing cells to evaluate the contribution of cell proliferation to wound healing in surgically injured tentacles of Electra pilosa. The data the authors present in this manuscript represent a significant contribution to our understanding of the bryozoan lophophore and animal cell structure. The methods the authors used are appropriate for their investigations, mostly clearly written, and sufficiently detailed for the work to be repeated. However, the authors may have biased the labeling of dividing cells to those cells with a relatively short cell cycle by using a labeling time of only six hours. For example, a labeling period of six hours is unlikely to detect cells that divide on the time scale of once every 3-4 days or once per week, particularly if the mitotic index of the cells in that region is low.

Validity of the findings

Findings on the organization and cell structure of the tentacles, lophophore base, and the ciliated pits

The authors provide a very good set of descriptions of the cellular structure of the tentacles, lophophore base, and ciliated pits for the four species they studied. Their observations on the cells of ciliated pit are of particular importance because this structure within the lophophore has been an understudied feature of bryozoans.

Findings on the regions of actively dividing cells in lophophores at different stages of development

The authors set out to collect data on cell division in the lophophore to evaluate “the hypothesis that tentacle growth/elongation is intercalary and cells’ proliferation takes place somewhere at the lophophore base because such pattern makes tentacle elongation possible without interrupting of the feeding process.” (lines 133-135).

In addition to the text from lines 133-135, the authors refer to the growth/elongation of the tentacles as being “intercalary”, see below. However, it is not clear to me what the authors are referring to as the hypothesis of intercalary tentacle growth.

Lines 34-37 This confirmed the suggested hypothesis about intercalary tentacle growth which provides a potential to alter a lophophore shape in adult polypides according to rearrangement of colony wide water currents during colony astogeny.

Lines 389-39: These patterns together with the absence of proliferating cells in the tentacles of adult polypides demonstrate early specialization of ciliated cells and provide evidence for intercalary growth of tentacles.

Lines 481-481 Recent study demonstrates that the specific design of bryozoan lophophore base enable intercalary growth of tentacles.

There are several other elements pertaining to the authors’ findings on dividing cells that require change or clarification:

a. The authors categorize the lophophores they examined as “young bud”, “late bud”, “young polypide”, and “adult or old polypide”. Young and late bud stages are developmental stages before the polypide is fully functional and depending on the size of the lophophore, I would expect that it is relatively easy to compare similar stages across the three species they examined. To stage young and adult polypides, the authors use zooid position within the colony relative to the growth margin. To identify these relative positions the authors use the term “generation”, with younger generations located closer to the growth margin. This terminology is confusing. First, the series of asexually budded zooids in a bryozoan colony are not usually referred to as different generations, because they represent clones of the same sexually produced ancestrula. Second, the numbering system is counter to what we normally think about in terms of generational ages (i.e., the 1st generation comes before the second generation). I would suggest that the authors consider just using the relative positions of zooids to the growth margin to estimate the “age class” of a polypide. In addition, the authors refer to some young polypides as “fully developed” which is not defined or distinguished from just young polypides (e.g., line 321).

b. A second concern is the actual “developmental age” of a lophophore relative to the position of its zooid in the colony. The authors state in lines 149-152 that “We numbered zooid generation starting from the colony edge and took into account only fully developed and actively feeding polypides, we did not consider buds.” I assume this means that the authors did not start their numbering system until they reached a point in the colony that was proximal to the growth margin and any zooids without an extended lophophore. The three species the authors examined have very different colony forms and I would expect very different rates of colony growth. For example, E. pilosa forms colonies that are lightly calcified, encrusting sheets; A. scabra forms colonies that are lightly calcified, biserial, and erect; and R. ovata forms colonies that are highly calcified and encrusting. The authors aging system may work well for comparisons within a colony or species, but it may be inappropriate for comparisons across the three species. For example, the distributions of EdU positive cells shown in figure 11 for young polypides of A. scabra and R. ovata are very different than that shown for a young polypide of E. pilosa. Is this a difference among species or age of lophophores? I recognize that the authors are doing their best to control for polypide age, but it might be helpful if the authors explicitly address this question as they describe their methods.

c. Variation in EdU-positive cells and Statistical analysis – on lines 324-327, the authors state for young fully developed polypides, “We registered from 8 up to 14 EdU-posistive cells in per ciliated pit. This parameter varies across ciliated pits and across polypides of the same generation for each studied species.” On lines 334-336, the authors state, “Though our study is qualitative and we did statistically analyze the proliferating activity at the lophophore base, the number of EdU-positive cells per ciliated pit in ‘old’ polypides is less than in ‘young’ ones (especially in E. pilosa): from 0 up to 4. Lastly, on lines 466-468, the authors state, “We did not statistically analyze the proliferating activity at the lophophore base since our study is qualitative and this is the first attempt to check the ability of tentacle regeneration in marine bryozoans.” I would agree with this last statement, that the authors did not actually statistically analyze the data. However, I think it is important that the authors provide at least some minimal additional information about the variation of EdU-positive cells in young and old polypides of each species, including sample size, mean, and some error term, such as standard deviation or standard error of the mean.

d. The authors present data that is clearly suggestive that cells of the ciliated pits and oral regions contribute to the growth of the tentacles. However, their data does not include definitive evidence that 1) EdU-labeled cells of the ciliated pits or oral region are incorporated into the tentacles or 2) the tentacles of the adult polypides they examined had grown (either increased in length or cell number) compared to the tentacles of the young polypides they examined. Consequently, the authors may want to be more careful in presenting and interpreting their data. For example:

Lines 337-338 – “The larger number of EdU-positive cells in ‘young’ polypides demonstrates that their tentacle still lengthen until their lophophores acquire a specific shape.”

Lines 397-399, “We are absolutely sure that CP- and DP-cells give C- and D-cells correspondingly since we traced stages of kinetosome biogenesis in these multiciliated cells.”

Lines 406-407 – “This study revealed the possible function of ciliated pits: they are responsible for tentacle elongation in young and adult polypides.”

e. The authors report a difference in the number of EdU-positive cells in the ciliated pits of young and adult lophophores (lines 324 and 336). However, it is not clear from the manuscript whether their description of ciliated pit organization and cell structure in lines 228-305 is based on studies of young and/or adult polypides. It would be interesting to know if the organization and/or cell structure of the ciliated pit differs between young and adult lophophores.

f. The authors demonstrate that during development, the cells of the lophophore that can be labeled with EdU in a six hour period become restricted to the ciliated pits and the base of the lophophore. The authors interpret regions of the lophophore without EdU-positive cells after a six hour labeling period as regions where cell division does not occur. However, extending the labeling period may detect slower dividing cells in other regions. Even the authors extended the labeling time to six hours to improve the EdU-positive signal in adult polypides (lines 311-312).

Findings on the contribution of cell proliferation to wound healing in surgically injured tentacles of Electra pilosa

The authors investigated if tentacles of E. pilosa regenerated after a piece of the distal end was surgically removed. The authors observed no regeneration over a seven-day period. Six hour incubations in EdU, labeled few cells in the tentacles, but did include an E-cell near the wounding site and a cell of the tentacle coelomic lining. The authors report an increase in EdU-positive cells in ciliated pits at the bases of the injured tentacles.

Additional comments

The caption for figure 12 does not identify the time after surgery for G. I believe the figures 12E-G are all 120 hours after surgery.

---

## Round 0.2 · Minor Revisions

The authors have improved their review according to the referees' comments although some minor revisions are still required (see reviewer and my comments below).

I suggest adding "almost exclusively colonial" or a similar sentence (line 82) as there are also solitary species (e.g. Schwaha et al. 2019: Aethozooides uraniae, a new deep-sea genus and species of solitary bryozoan from the Mediterranean Sea, with a revision of the Aethozoidae).

Please add missing data of some species: author(s) of the scientific name and the year it was published when a species' name is cited in the manuscript for the first time (e.g. Cryptosula pallasiana (Moll, 1803) line 112).

Reviewer 1 ·

Basic reporting

The text has much improved compared to the previous version. The authors have done a very good job incorporating most of the reviewers' comments. The language has improved. Although there are still few minor shortcomings, I am confident these can easily be addressed by a thorough typesetting. I have no further concerns or objections and can recommend publication of this manuscript. All positive comments given in my initial review still apply.

Some minor typos/errors I have picked up on the way:
124 ‘predate’ should be changed to ‘prey’
133 ‘basing’ should read ‘based’
168 ‘was’ should be ‘were’
374 ‘that’ should be ‘those’
475 I think you mean your study? Then change ‘A recent’ to ‘This’. Otherwise provide reference…
502 same as the above comment

Experimental design

OK, as previously reported

Validity of the findings

Also still OK, comments in this direction made by reviewer 3 and myself have adequately been dealt with

---

## Round 0.3 · accepted · Accept

Dear Dr. Shunatova,

I am pleased to inform you that your paper has been accepted for publication without further changes.

Thank you for submitting your work to PeerJ. We hope you consider us again for future submissions.

Best regards,

Blanca Figuerola
Academic Editor, PeerJ